# Long-term deglacial permafrost carbon dynamics in MPI-ESM

Thomas Schneider von Deimling[1,2], Thomas Kleinen[1], Gustaf Hugelius[3,4], Christian Knoblauch[5], Christian Beer[6,4], Victor Brovkin[1]

[1]Max Planck Institute for Meteorology, Bundesstr. 53, 20146 Hamburg, Germany
[2] now at Alfred Wegener Institute Helmholtz Centre for Polar and Marine Research, 14473 Potsdam, Germany
[3]Department of Physical Geography, Stockholm University, SE10693, Stockholm, Sweden
[4]Bolin Climate Research Centre, Stockholm University, SE10693, Stockholm, Sweden
[5]Institute of Soil Science, Universität Hamburg, Allende-Platz 2, 20146 Hamburg, Germany
[6]Department of Environmental Science and Analytical Chemistry, Stockholm University, 10691 Stockholm, Sweden

*Correspondence to*: Thomas Schneider von Deimling (thomas.schneider@awi.de)

## Abstract

We have developed a new module to calculate soil organic carbon (SOC) accumulation in perennially frozen ground in the land surface model JSBACH. Running this offline version of MPI-ESM we have modelled long-term permafrost carbon
accumulation and release from the Last Glacial Maximum (LGM) to the Pre-industrial (PI). Our simulated near-surface PI permafrost extent of 16.9 Mio km$^2$ is close to observational estimates. Glacial boundary conditions, especially ice sheet coverage, result in profoundly different spatial patterns of glacial permafrost extent. Deglacial warming leads to large-scale changes in soil temperatures, manifested in permafrost disappearance in southerly regions, and permafrost aggregation in formerly glaciated grid cells. In contrast to the large spatial shift in simulated permafrost occurrence, we infer an only
moderate increase of total LGM permafrost area (18.3 Mio km$^2$) – together with pronounced changes in the depth of seasonal thaw. Earlier empirical reconstructions suggest a larger spread of permafrost towards more southerly regions under glacial conditions, but with a highly uncertain extent of non-continuous permafrost.

Compared to a control simulation without describing the transport of SOC into perennially frozen ground, the implementation of our newly developed module for simulating permafrost SOC accumulation leads to a doubling of
simulated LGM permafrost SOC storage (amounting to a total of ~150 PgC). Despite LGM temperatures favouring a larger permafrost extent, simulated cold glacial temperatures – together with low precipitation and low $CO_2$ levels – limit vegetation productivity and therefore prevent a larger glacial SOC build-up in our model. Changes in physical and biogeochemical boundary conditions during deglacial warming lead to an increase in mineral SOC storage towards the Holocene (168 PgC at PI), which is below observational estimates (575 PgC in continuous and discontinuous permafrost).
Additional model experiments clarified the sensitivity of simulated SOC storage to model parameters, affecting long-term soil carbon respiration rates and simulated active layer depths. Rather than a steady increase in carbon release from the LGM to PI as a consequence of deglacial permafrost degradation, our results suggest alternating phases of soil carbon

accumulation and loss as an effect of dynamic changes in permafrost extent, active layer depths, soil litter input, and heterotrophic respiration.

## 1    Introduction

The amount of carbon stored in the atmosphere, in the ocean, and on land has varied strongly between glacial and modern times (Ciais et al., 2012). Ice-core records suggest a large increase of about 100 ppm in atmospheric $CO_2$ concentrations from the Last Glacial Maximum (LGM) to the pre-industrial (PI) climate, posing the question of the source of this atmospheric carbon input. Given the overwhelmingly large storage capacity of the global oceans, the release of oceanic $CO_2$ played a dominant role in the atmospheric $CO_2$ increase during deglaciation (Archer et al., 2000;Brovkin et al., 2012). There are especially large uncertainties in estimates of past terrestrial carbon storage dynamics. During deglacial warming from the LGM to the PI climate, global land vegetation was a strong sink of carbon through increased productivity stimulated by warmer temperatures and higher $CO_2$ levels. A key question concerns the role of terrestrial soils along the transition from the Glacial to the Holocene with regard to their acting as a carbon source or a carbon sink. In this study we use the Earth System Model MPI-ESM to investigate soil organic carbon (SOC) accumulated in permafrost under glacial conditions, and the dynamics of carbon uptake and release under deglacial warming into the PI climate.

Large amounts of SOC have accumulated in soils of the northern permafrost regions as a consequence of the low heterotrophic respiration in the surface soil layer that thaws during the short summer period (active layer) and permanent sub-zero temperatures in permafrost. Vertical soil mixing through consecutive freeze-thaw cycles (cryoturbation) is specific to permafrost soils and further favours high SOC accumulation, but is generally not accounted for in current Earth System Models (Koven et al., 2009;Koven et al., 2015;Beer, 2016). The large carbon storage capacity of high latitude soils is underlined by observational evidence which points to a total SOC stored in permafrost region of ~1300 PgC under present-day climate conditions (Hugelius et al., 2014). This large carbon pool represents a major part of global SOC storage (Jackson et al., 2017) and is therefore considered an important component in the global carbon cycle. Model projections of future permafrost degradation and consequent carbon release have underlined the vulnerability of the permafrost carbon store to warming and have discussed implications for affecting future greenhouse gas levels (McGuire et al., 2009;Lawrence et al., 2011;Koven et al., 2011;Schneider von Deimling et al., 2012;Burke et al., 2012;Schaphoff et al., 2013;Schaefer et al., 2014). Given the sensitivity of permafrost  soils to climate change, manifested in permafrost aggregation or degradation, which decreases or increases $CO_2$ and $CH_4$ release from permafrost soils, permafrost carbon is also discussed in the paleo-climatic context. It has been suggested that the amount of permafrost SOC has been distinctively different under glacial and modern climate conditions (Ciais et al., 2012). Zimov et al.(2006) speculate that the thawing of frozen loess in Europe and southern Siberia has led to large carbon releases during the Pleistocene to Holocene transition. Isotopic analyses of ice-core $CO_2$ have pointed to large excursions of [13]C-depleted carbon, interpreted as a possible strong land source from permafrost carbon release (Lourantou et al., 2010). Recently Crichton et al. (2016) speculated that permafrost carbon release likely played a

dominant role for explaining a pronounced $CO_2$ rise between 17.5 to 15 kyrs BP, while Köhler et al. (2014) have analysed $\Delta^{14}C$ excursions from coral records to explain abrupt release of about 125 PgC from permafrost degradation at the onset of the Bølling-Allerød (~14.600 kyrs BP). A recent empirically-based reconstruction suggests that the total estimated SOC stock for the LGM northern permafrost region is smaller than the present-day SOC storage for the same region (Lindgren et al., 2018). This reconstruction shows that very significant changes in SOC have occurred over this time interval, including decreased Yedoma SOC stocks and increased peat SOC stocks, but does not shed light on when these shifts occurred (Lindgren et al., 2018).

A focus of many paleo modelling studies is the LGM climate, as it was distinctively different to the PI climate and therefore is considered an ideal test case for model evaluation, e.g. within the framework of paleo model intercomparisons (PMIP, (Braconnot et al., 2012;Kageyama et al., 2017). With respect to permafrost dynamics, PMIP models have been used to analyse permafrost extent under LGM and PI climate conditions (Saito et al., 2013), but not with respect to changes in permafrost soil carbon storage. Willeit et al. (2015) have run the Earth system model CLIMBER-2 over the last glacial cycle to explore the interaction of permafrost and ice-sheet evolution. Using the ORCHIDEE land surface model, Zhu et al. (2016) simulated glacial SOC storage in Yedoma deposits showing a rather large potential of these thick ice- and organic-rich sediments to affect the global carbon cycle. Having analysed a permafrost loess-paleosol sequence, Zech et al. (2011) inferred that more soil organic carbon was sequestered during glacials than during interglacials.

Permafrost carbon dynamics are only recently being implemented into Earth-System-Models (ESMs). Due to this, and the computational costs of high-resolution glacial-interglacial model runs, we know of no previous full-complexity ESM experiments (in contrast to EMIC studies) that quantify the dynamic role of circum-Arctic permafrost carbon during deglacial warming.

In this study, we investigate permafrost soil carbon dynamics from the LGM to PI using a process-based land surface model. In particular, we want to analyse the amount of SOC which was stored under LGM climate conditions, and the dynamics of this SOC store under deglacial warming into the Holocene – in response to receding ice-sheets, broad-scale shifts in permafrost regime, and increases in vegetation productivity and soil respiration.

## 2    Model description

### 2.1    Simulating deglacial climate dynamics

For simulating soil carbon dynamics from the LGM to PI climate, we have used a standalone (offline) configuration of the MPI-ESM land surface model JSBACH (Reick et al., 2013;Brovkin et al., 2013;Schneck et al., 2013) in coarse T31GR30 resolution (approximately 3.75° or roughly 400 km x 300 km at 45°N). The model was run with dynamic vegetation and driven by transient climate forcings covering the full LGM to PI period.  In contrast to the CMIP5 version previously published, we use an extended version of JSBACH with a multilayer hydrology scheme (Hagemann and Stacke, 2015), a representation of physical permafrost processes (Ekici et al., 2014), and an improved soil carbon model based on the YASSO

model (Goll et al., 2015). Climate forcing fields were prescribed for surface air temperature, precipitation, humidity, radiation and wind speed and were inferred from a LGM time-slice experiment with MPI-ESM1.2 without permafrost physics, combined with a full transient glacial cycle experiment performed with the ESM of intermediate complexity CLIMBER2 (Ganopolski et al., 2010). In Appendix 7.1 we describe in detail the generation of the climate driving fields

applied to JSBACH and further show the performance of MPI-ESM1.2 of simulating pre-industrial and glacial temperature and precipitation fields in high northern latitudes.

## 2.2    Model set-up

In this study we focus on long-term (millennial scale) dynamics of near-surface SOC stocks, and therefore only consider carbon accumulation in permafrost soils to a depth of three meters. Consequences of accounting for deeper SOC deposits are

briefly discussed in section 2.5. For many JSBACH permafrost grid cells, bedrock depths are shallower than three meters and therefore limit the maximum depth of SOC accumulation in our model. Here, we use a global dataset of soil depths compiled by Carvalhais (2014), see Figure A8). Vegetation cover is assumed to respond to changing glacial-Holocene climatic conditions and is calculated dynamically throughout the simulation. Pre-industrial land use changes in high latitudes are negligible (Hurtt et al., 2011), and therefore we do not account for human-induced modifications of vegetation cover. We

prescribe glacial ice sheet coverage and $CO_2$ concentration (see section 7.4) along with its changes during deglacial warming based on the CLIMBER2 model (Ganopolski et al., 2010). We do not consider glacial lowering of sea-level and do not account for permafrost having established in shelf regions which were exposed to cold surface air conditions under a lower glacial sea level.

## 2.3    Physical permafrost model

Here we describe key aspects of the physical soil representation in JSBACH, while a detailed model description can be found in Ekici et al. (2014). In our study we use a setup with eleven vertical soil layers of increasing vertical thicknesses reaching a lower boundary at 40 meters. Surface air temperature is used as an upper boundary forcing for calculating soil temperatures during the snow-free season. When snow is present, a five-layer snow scheme is applied for forcing soil temperatures. An organic layer of constant thickness is assumed to cover the soil top. The bottom boundary condition is

given by a zero heat flux assumption. Heat transfer into the soil is calculated for each soil layer by using a one-dimensional heat-conduction equation accounting for phase change of soil water.

## 2.4    Soil carbon model

Soil carbon dynamics within JSBACH are simulated by YASSO (Liski et al., 2005). We describe key characteristics of this soil carbon model, while a detailed description of its implementation into JSBACH can be found elsewhere (Thum et al.,

2011;Goll et al., 2015). YASSO calculates the decomposition of soil organic matter considering four different lability classes based on a chemical compound separation of litter into acid-soluble (A), water-soluble (W), ethanol-soluble (E) and non-

soluble (N) fractions with the pools replicated for above- and belowground. In addition to these four pools, there is a slow pool that receives a fraction of the decomposition products of the more labile pools (Fig.1). These pools are replicated for green and woody litter on each vegetation tile. Decomposition rates of each carbon pool were inferred from litter bag experiments and soil carbon measurements and range from turnover times of up to few years (labile pools) to multi-

centennial for the slow pool. Furthermore, litter input into the soil is considered separately for leaf and woody components (not shown in Fig.1), assuming decreasing decomposition rates for increasing size of woody litter. The dependence of decomposition on temperature is described by an optimum curve, combined with a scaling factor $k(precip) = 1 - e^{-1.2*precip}$ to describe precipitation dependency (Tuomi et al., 2009). YASSO decomposition parameters were tuned to surface air temperatures and precipitation, using measurements that include sites representing tundra conditions. With

increasing active layer depths, temperatures above the permafrost are much colder than surface temperatures during summer – and therefore suggest lower decomposition rates compared to the soil surface. Although the temperature difference between soil and surface air temperature is growing with depth, the typical vertical profile of declining SOC (section 7.7) introduces less weight to deeper layers. Given that the more labile SOC pools are concentrated in upper soil layers (Walz et al., 2017), the soil-surface air temperature difference is mainly affecting carbon pools of slow decomposition (N and H pool,

section 7.7). In section 4.3 we investigate implications of assuming a reduced decomposition timescale for the slow H pool on SOC build-up. A consideration of decomposition parameters depending on soil temperature profiles is subject to current development work for JSBACH.

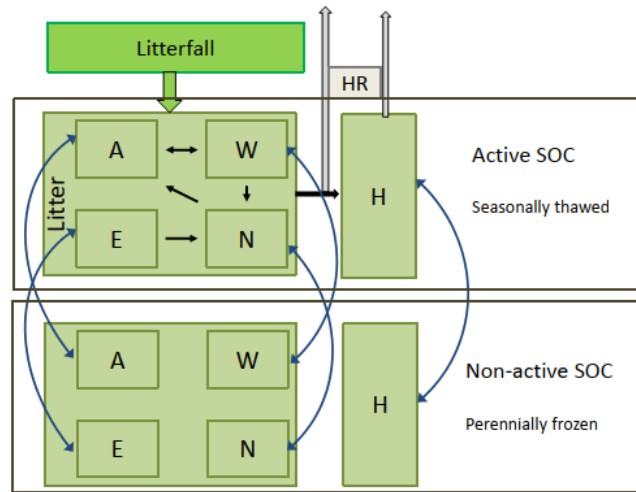

**Figure 1: Extended soil carbon pool structure in JSBACH accounting for active layer and permafrost carbon (modified from Goll**

**et al. 2014). In YASSO, soil organic matter is separated into groups of different chemical compounds (A,W,E: labile pools), an**

**intermediate pool (N), and a slow pool (H). Carbon gain results from litter input, carbon loss from heterotrophic respiration (HR) in active layer pools. Changes in maximum seasonal thaw depth induce a transfer of carbon (blue arrows) from non-active to active pools (warming), and vice versa (cooling).**

A main limitation for modelling the transfer of carbon between the active layer and the underlying permafrost body is the zero-dimensional structure of YASSO within JSBACH which does not allow calculating vertical SOC profiles. Yet, permafrost-affected grounds can store SOC in depths of several meters in the soil and reveal rather pronounced vertical gradients of typically decreasing carbon concentrations with depth within the active layer (Harden et al., 2012). Therefore we have developed a soil carbon build-up model which describes carbon gain through litter input, carbon loss through heterotrophic respiration, and vertical carbon transport through cryoturbation and sedimentation. Based on this model, we infer the needed information of vertical SOC distributions (see Appendix). We then use this new model to determine SOC concentrations at the level of the active layer depth (ALD) which determine the amount of carbon transferred between perennially-frozen (permafrost) and seasonally-thawed (active layer) pools. For this purpose we added in JSBACH for each YASSO soil carbon pool an additional non-active pool which describes carbon of the same chemical compound class, but which is not subject to seasonal thaw and remains perennially frozen (Fig.1).

For each grid cell and each soil organic matter (SOM) class $i$ the carbon transfer $dSOC^i(t)$ at time-step $t$ (in kgC m$^{-2}$) between thawed (active) and perennially frozen (non-active) pools is simulated in JSBACH depending on the extent of annual change in ALD (in meter), and on the individual SOC concentration $SOCC^i_{ALD}$ (in kgC m$^{-3}$) at the boundary between active layer and permafrost SOC :

$$dSOC^i(t) = dALD(t) * SOCC^i_{ALD}(t) \tag{1}$$

with $dALD(t)$ describing the change in maximum thaw depth. As our focus is on long-term carbon transfer we smooth $dALD(t)$ by applying a 100 year exponential weighting. $SOCC^i_{ALD}(t)$ is the soil organic carbon concentration for each lability class $i$ at the boundary between seasonally thawed and perennially frozen ground.

The transfer of carbon into permafrost depends strongly on the thickness of the active layer: with increasing thickness, the share of labile soil organic material, which gets incorporated into permafrost, decreases as the distance of carbon transport to the permafrost boundary gets longer and therefore allows for more time of microbial decomposition. Our approach enables us to capture this key characteristic of soil carbon accumulation in permafrost soils, i.e. it describes the fractionation of SOM of differing lability with depth. As a consequence, shares of more labile SOC are increasing when organic material accumulates in colder climate conditions under shallower active layer depths (Figure A11).

## 2.5    Model limitations

The full dynamics of deglacial permafrost SOC accumulation and release are determined by a multitude of factors, ranging from past climatic boundary conditions and associated permafrost evolution to process-based descriptions of SOC formation. Here we discuss structural model aspects and model limitations with regard to these factors which can explain part of the model data discrepancies which we discuss in section 4.

### 2.5.1    Unaccounted aspects of permafrost extent and dynamics

The T31 resolution of JSBACH used in this study has allowed us to run a set of model experiments from the LGM to the PI. Yet, the coarse resolution does not allow simulating permafrost which covers only a fraction of the landscape (such as isolated and sporadic permafrost). We therefore underestimate the total area of ground subject to perennially frozen conditions.

We also do not account for effects of excess-ice which affects the temporal thaw dynamics and soil moisture conditions upon thaw in ice-rich grounds (Lee et al., 2014).

### 2.5.2    Unaccounted permafrost carbon stocks

#### a)    Peatlands

We do not account for water-logged soils – an environment which allows the formation of peatlands and which requires specific model descriptions of SOC build-up (Kleinen et al., 2012). As a peat module was not included in our version of MPI-ESM, we do not describe carbon storage following deglacial peat dynamics (Yu et al., 2010). We focus in this study on typical carbon profiles in mineral soils which decline with depth (in line with large-scale observational evidence, (Harden et al., 2012). We acknowledge that we therefore underestimate total carbon storage by not capturing high carbon accumulation in saturated, organic rich soils (see discussion in section 4.1.4).

#### b)    Deep SOC deposits

We do not account for the evolution of syngenetic permafrost deposits through sustained accumulation of new material on the top of the active layer. High sedimentation rates result in deep soil deposits, rich in organic matter, as found in Yedoma soils or river deltas, which can store hundreds Pg of SOC down to some tens of meters (Hugelius et al., 2014;Strauss et al., 2017). Using a multi-box model of permafrost carbon inventories, (Schneider von Deimling et al., 2015) have shown that large amounts of perennially frozen carbon in depths below three meter can get mobilized on a century timescale if abrupt thaw through thermokarst lake formation is accounted for. A consideration of the accumulation and release dynamics of these deep deposits under deglacial warming in JSBACH would require a process-based description of Yedoma formation and of abrupt thaw processes which is beyond the scope of this study.

### 2.5.3    Simulation of SOC respiration loss

Our model of describing soil carbon dynamics (YASSO) is based on the assumption that soil carbon decomposition can be inferred based on a chemical compound separation of litter. Model parameters are fitted based on annual litter-bag experiments. Therefore, uncertainties in long-term soil carbon dynamics (centennial to millennial timescale) can be large. Other factors, including a strong dominance of roots, rather than surface litter, in the contribution to soil OM, soil texture and mineralogy can strongly affect decomposition timescales. The stabilization of SOM due to its interaction with mineral compounds strongly reduces SOM decomposability (Schädel et al., 2014;Xu et al., 2016), but this particular process is not considered in YASSO.

Probably of more importance is our omission in the process-based SOM transport model of not explicitly accounting for soil moisture effects on SOC build-up which e.g. can result in profiles of increasing SOC with depth (Zimov et al., 2006) due to slowing down respiration with soil moisture approaching saturation (see also discussion in Appendix 7.7.1).

We further focus on SOM decomposition under aerobic conditions and do not model $CH_4$ formation and release in anaerobic soil environments.

### 2.5.4    Stationary assumptions

a)   Soil depths

We do not model soil genesis but prescribe stationary soil depths according to Carvalhais et al (2014) by assuming a balance between deglacial soil erosion and formation rates. In Appendix 7.4.3 we discuss implications of this assumption on modelled SOC build-up. We account for the fact that many soils have only formed after the LGM by assuming that SOC-build-up was prevented for grid cells when covered by LGM ice sheets.

b)   Organic surface layer

Protection of warm permafrost through insulation effects by organic surface layers have likely varied between the glacial and Holocene climate but are not captured in our modelling study. A more elaborate scheme of organic layer treatment is subject to current JSBACH model development, coupling organic layer thickness to litter carbon amounts.

c)   Vertical SOC gradients

Our approach of accounting for vertical SOC gradients (section 7.7) is based on using a process-based model of SOM transport assuming equilibrium conditions. Transient deviations from the equilibrium profiles cannot be captured and would be most pronounced for the more fast cycling SOC pools, which reveal pronounced vertical gradients (Figure A11()). In this study we focus on long-term carbon dynamics along the deglacial warming, and therefore do not capture full SOC dynamics resulting from short-term climate changes on decadal to centennial timescales.

## 3      Simulation design

### 3.1      Simulation set-up

For spinning-up soil carbon pools we start our simulations at 28 ka before present (BP) from zero soil carbon concentrations and run JSBACH for seven thousand years under a stationary climate forcing representative for LGM conditions. The chosen

spin-up time allows the slow soil carbon pools to come close to equilibrium (see Figure A 13). As cold glacial climate conditions have prevailed for many ten thousands of years before LGM, we assume that soils had enough time to accumulate soil carbon into permafrost under pre-LGM conditions, leading to approximately depth-constant SOC profiles between the permafrost table and our considered lower soil boundary (see Figure A11 and discussion in Appendix 7.7.1). With a focus on near-surface permafrost in this study, this lower boundary $z^*$ for SOC accumulation is assumed at three meters, while the

lower boundary for modelling soil temperatures is at 40 meters. More shallow SOC accumulation is assumed if soil depth is constrained by shallow laying rock sediments which we prescribe after Carvalhais et al. (2014, see Figure A8). In Appendix 7.4.3 we discuss implications of prescribed soil depths on SOC build-up.

Further, we assume that soil organic matter accumulation is prevented for sites covered for millennia under LGM ice sheets.

During the LGM spin-up phase we calculate $SOC_{PF}^i$ accumulation in perennially frozen ground (in kgC m$^{-2}$) for ice-free sites for each SOM lability class $i$ (following YASSO) as:

$$SOC_{PF}^i(t) = SOCC_{ALD}^i(t) * \left(z^* - ALD(t)\right), \tag{2}$$

with soil carbon concentrations $SOCC_{ALD}^i(t)$ at the permafrost table (in kgC m$^{-3}$) inferred as described in Appendix 7.7, and $z^*$ describing the lower boundary of SOC accumulation (in meter).

After 7000 years of model spin-up (when we diagnose permafrost SOC), we activate our scheme of transient SOC transfer between seasonally-thawed and perennially-frozen SOC pools and calculate $SOC_{PF}^i$ prognostically with the transfer rate depending on changes of simulated active layer depth in each grid cell (see Appendix 7.7). We run the model for another

1000 years under stationary LGM climate forcing until 20 ka BP to allow for equilibration of the transient SOC transfer scheme and then perform the fully-transient deglacial simulation from 20ka BP to PI (see Figure A 13).

### 3.2      Model experiments

Unless otherwise stated, we discuss simulated model output by referring to our transient model experiment from LGM to PI

with standard model parameters (experiment L2P). The model requires 16.43s per model year on 108 nodes of our high-

performance machine, giving a total computation time requirement of 0.5 node-h/yr. To evaluate uncertainty in parameter settings (see

Table 1), we have performed a set of additional experiments with JSBACH which are described in the following subsections.

| Experiment label | Experiment description | Configuration | Experiment setting |
|---|---|---|---|
| **L2P** | Reference run LGM to PI | Uncoupled atmosphere-ocean | Transient run LGM to PI<br>Standard model parameters<br>Prescription of glacial boundary conditions |
| **L2P_ALD** | Decreased active layer depth | Uncoupled atmosphere-ocean | Transient run LGM to PI<br>Thermal conductivity of soil surface organic layer reduced by factor 2 (i.e. 0.125 W m-1 K-1) |
| **L2P_VMR** | Increased vertical SOC mixing | Uncoupled atmosphere-ocean | Transient run LGM to PI<br>Cryoturbation rate of process-based SOM transport model increased by factor 2 (i.e. 20 cm$^2$ yr$^{-1}$)<br>Modified fit parameters for describing $SOCC_{ALD}/SOCC$ |
| **L2P_HDT** | Decreased decomposition of the slow pool | Uncoupled atmosphere-ocean | Transient run LGM to PI<br>Slow pool decomposition timescale parameter increased from 625 to 1000 years |
| **L2P_LIT** | Increased litter input | Uncoupled atmosphere-ocean | Transient run LGM to PI<br>Doubled litter input to YASSO soil model |
| **L2P_CTR** | Control run | Uncoupled atmosphere-ocean | Transient run LGM to PI<br>Standard model parameters, but without SOC transfer to permafrost |
| **MPI-ESM1.2 T31_PI** | PI time slice | Coupled atmosphere-ocean | Pre-industrial run without permafrost physics in T31GR30 resolution (corresponding to ~ 400 km x 300 km at 45°N) |
| **MPI-ESM1.2 T31_LGM** | LGM time slice | Coupled atmosphere-ocean | Last Glacial Maximum run without permafrost physics in T31GR30 resolution (corresponding to ~ 400 km x 300 km at 45°N) |

**Table 1: Performed transient and time slice model experiments with MPI-ESM**

### 3.2.1   Active layer depth (experiment L2P_ALD)

Organic layers at the top of permafrost soils cover only a small fraction of the soil profile, but exert a large effect on subsoil-
10   temperatures by their insulating effect (Porada et al., 2016;O'Donnell et al., 2009). Given a tendency of overestimating active layer depths in JSBACH compared to observations (see section 4.1.2), we have set up a sensitivity experiment in which we have lowered the standard value of thermal conductivity of the surface organic layer (0.25 W m$^{-1}$ K$^{-1}$, (Ekici et al., 2014) by a factor of two to test the significance of this parameter.

### 3.2.2 Vertical mixing rates (experiment L2P_VMR)

Vertical mixing of SOC in permafrost soils through cryoturbation is a well-studied process, but as the timescale involved is rather large, the magnitude of cryoturbation mixing rates is hard to constrain by observations and is subject to large uncertainty (Koven et al., 2009). We investigate consequences of doubling our assumed standard cryoturbation rate of 10

$cm^2$ $yr^{-1}$. Unless high sedimentation regions of fast loess formation are considered (e.g. (Zimov et al., 2006), sedimentation has a much less pronounced effect on vertical concentration profiles of SOC in our model setting than cryoturbation and is therefore not considered separately.

### 3.2.3 Slow pool decomposition timescale (experiment L2P_HDT)

Decomposition timescale parameters in YASSO are inferred from a large set of litter-bag experiments. However, the slow

pool decomposition timescale is less constrained. We therefore perform an additional experiment in which we increase the standard reference slow pool turnover time of 625 years (assumed for 0 °C and unlimited water availability) to 1000 years.

### 3.2.4 Litter input (experiment L2P_LIT)

SOC accumulation depends critically on simulated litter input. In our standard simulation we simulate a rather low NPP in permafrost regions under the harsh climatic conditions at 20 ka BP (see discussion in section 4.1.3). We therefore have

performed an additional experiment in which we have doubled litter input to YASSO soil pools to investigate SOC accumulation under a more productive LGM vegetation compared to our standard parameter setting.

### 3.2.5 Pre-Industrial time-slice (MPI-ESM1.2 T31_PI)

Time-slice experiment, in which MPI-ESM was run without permafrost physics in T31GR30 resolution in a fully-coupled atmosphere-ocean setting (see section 7.2). Stationary PI boundary conditions were defined following the CMIP5 protocol.

### 3.2.6 LGM time-slice (MPI-ESM1.2 T31_LGM)

Time-slice experiment for stationary LGM boundary conditions following the PMIP3 protocol (Braconnot et al., 2011), with LGM land-sea and ice sheet masks, as well as greenhouse gases and orbit modified to LGM conditions (see section7.3). The model was run without permafrost physics in a fully-coupled atmosphere-ocean setting in T31GR30 resolution.

## 4   Results and Discussion

We first show simulated spatial patterns of physical and biogeochemical drivers for SOC build-up, along with simulated SOC distributions in permafrost regions under LGM and PI climatic conditions. We then discuss transient SOC dynamics

from the LGM at 20 ka BP to the Holocene at 0 BP. Finally, we discuss the robustness of our findings with regard to uncertainty in specific model parameter choices.

## 4.1 PI and LGM time slices

### 4.1.1 Permafrost extent at PI and LGM

Under PI climate conditions we model a northern hemisphere permafrost extent of 20.3 million $km^2$ (Figure 2). Hereby we classify a grid cell subject to permafrost if maximum seasonal thaw is consistently less than the model's lowest soil boundary at 40 meters. Most of the areal coverage is in Asia where the southern boundary extents to 46 °N, excluding more southerly permafrost in the Himalaya region (not shown). Focusing on near-surface permafrost within the upper three meters of the soils, JSBACH simulates a northern permafrost extent of 16.9 million $km^2$. Data-based estimates indicate that about

24% of the exposed northern hemisphere (NH) land area or 22.8 million $km^2$ are affected by permafrost (Zhang et al., 1999). These estimates comprise permafrost regions of smaller-scale occurrence, such as *sporadic* permafrost (10-50% landscape coverage) or *isolated* permafrost (less than 10% coverage). These smaller landscape-scale features are not captured by JSBACH grid cell sizes. When excluding these contributions, data-based estimates of continuous and discontinuous permafrost suggest an areal coverage of 15.1 million $km^2$ (Zhang et al., 2008), about 90% of JSBACH simulated near-

surface pre-industrial permafrost extent.

Under the cold climatic conditions prevailing at LGM, JSBACH simulates an additional 3.7 million $km^2$ above pre-industrial extent, amounting to a total NH permafrost area of 24 million $km^2$, which is close to the mean of PMIP3 model results of 26 million $km^2$ (Saito et al., 2013). Our simulated LGM near-surface permafrost in the upper three meters amounts to 18.3 million $km^2$. Reconstructions suggest a total coverage of about 30 million $km^2$ on current land areas (Lindgren et al., 2016).

Without contributions from discontinuous permafrost, the reconstructed extent (25.4 $km^2$) is close to our simulated total LGM permafrost extent. Compared to the reconstructions, JSBACH simulates less LGM permafrost in Europe, western and central Asia, and slightly more permafrost in North America at the southern Laurentide ice sheet boundary. The discrepancy between the model and data is likely a consequence of too warm soil temperatures simulated under LGM conditions at southerly permafrost grid cells. Part of the discrepancy might also be explained if LGM data-based estimates of

discontinuous permafrost comprise large contributions from sporadic and isolated permafrost not resolved in JSBACH.

Another part of the discrepancy might result from precipitation biases leading to snow depth biases under glacial conditions. Overestimates in simulated snow depth can easily translate into too excessive snow insulation and therefore result in unrealistic high soil temperature (Stieglitz et al., 2003;Zhang, 2005;Lawrence and Slater, 2010;Slater and Lawrence, 2013;Langer et al., 2013).

Simulating too little LGM permafrost coverage underlines the challenge of modelling warm permafrost occurrence in non-continuous permafrost regions in which smaller-scale variations in snow thickness, vegetation cover, and topography play an increasingly dominant role on the soil thermal state.

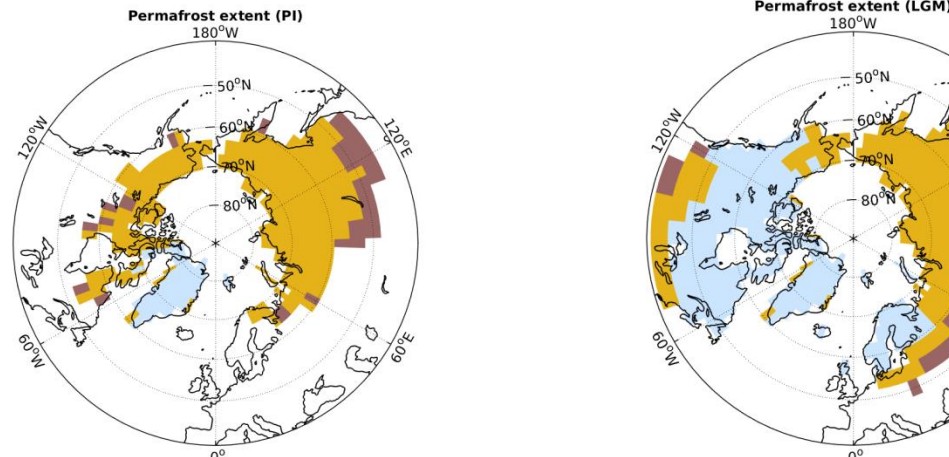

**Figure 2: Ice sheet coverage and permafrost extent at PI and LGM simulated with JSBACH. Mustard areas illustrate grid cells of simulated near-surface permafrost with active layers above three meters (PF$_{3m}$), brown-mauve areas describe permafrost with active layers deeper than three meters (PF). Light bluish areas show prescribed ice sheet coverage (IS). Data shown represent hundred year time averages.**

Despite a limited simulated decrease in permafrost extent (20%), pronounced spatial changes in permafrost coverage from the LGM to the PI climate are evident (Figure 2). As a consequence of the cold climatic conditions prevailing at the LGM, permafrost extent has spread further south in most regions. At the same time, large areas of the northern hemisphere, especially in North America have been covered by thick ice sheets, thus limiting the maximum area for permafrost to establish in northern hemisphere grounds.

### 4.1.2    ALD for PI and LGM

Figure 3 shows simulated active layer depths for PI and glacial conditions. Compared to the PI experiment performed with the CMIP-5 MPI-ESM model version, a slight warm bias in simulated mean surface air temperatures is evident in most grid cells of North America for MPI-ESM1.2T31 (Figure A2). As a consequence of this warm bias, active layer depths in Alaska are biased high for most grid cells (Figure A9) in our model version when compared to CALM observations (Brown et al., 2000). Given rather poor data coverage of monitored active layer depths especially in Asia, a model-data comparison should be seen with caution as it is questionable to what extent the site-level data are representative for scales simulated by JSBACH. Nevertheless, we here compare large-scale simulated ALDs to local-scale observational estimates based on the CALM database. Model data mismatches are less pronounced in Asia, and generally the model experiment of an increased organic layer insulating effect (experiment L2P_ALD, see section 4.3) suggests improved agreement with the data. The

simulated active layer depths further suggest a tendency of too warm soil temperatures in southerly permafrost regions (see Figure A9). This is in line with underestimating LGM southward spread of permafrost in JSBACH compared to reconstructions.

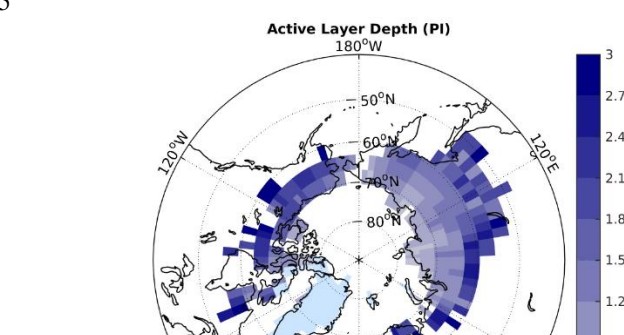 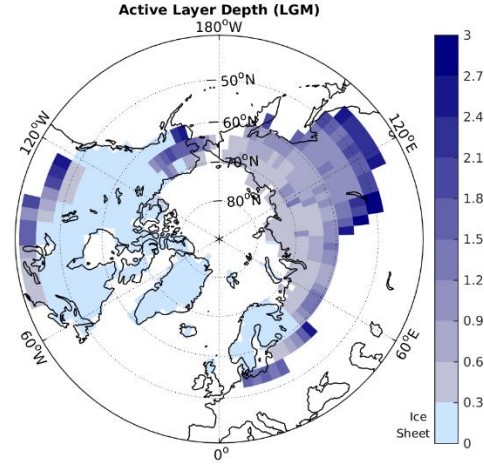

**Figure 3: Active layer depths in near-surface permafrost soils at PI and LGM simulated by JSBACH. Light bluish areas show prescribed ice sheet coverage. Data shown represent hundred year time averages.**

### 4.1.3    Vegetation productivity at PI and LGM

The amount of SOC stored in the ground depends crucially on vegetation litter input. Figure 4 shows high latitude vegetation productivity under LGM and PI climate conditions. We infer highest vegetation productivity (with NPP larger than 250 gC m$^{-2}$ yr$^{-1}$) for the PI at the southern near-surface boundary in North America. The harsh glacial conditions of low temperatures

15    and low precipitation in combination with low $CO_2$ levels result in strongly reduced glacial vegetation productivity compared to the PI. Therefore large regions show a NPP well below 50 gC m$^{-2}$ yr$^{-1}$ during the LGM (Figure 4). Especially the sensitivity of vegetation productivity to changing $CO_2$ levels is debatable. This structural model uncertainty is underlined by an ensemble of process-based land-surface models which reveal a very large spread in simulated vegetation productivity to changing $CO_2$ levels under present-day climate (McGuire et al., 2016). Beer et al. (2010) have inferred observation-based

20    estimates of GPP and have underlined the large uncertainty in modelled high latitude vegetation. Compared to this study, our simulated NPP falls in the lower range of their estimates. For a more detailed model-data comparison of vegetation productivity we have analysed up-scaled flux tower measurements of GPP (Jung et al., 2011) by assuming that present-day GPP is roughly 15% above pre-industrial values (Ciais et al., 2013).  In Figure A4 we analyse simulated pre-industrial GPP and infer a larger vegetation productivity in North America (about 25-50% above the data), while a reversed signal of low-

biased GPP in Eurasia (typically about 50% below the data) is modelled. This pattern of GPP deviation from observations expresses the climatology bias of this model configuration of MPI-ESM1.2 due to an altered land-atmosphere coupling (see discussion in section 7.2).

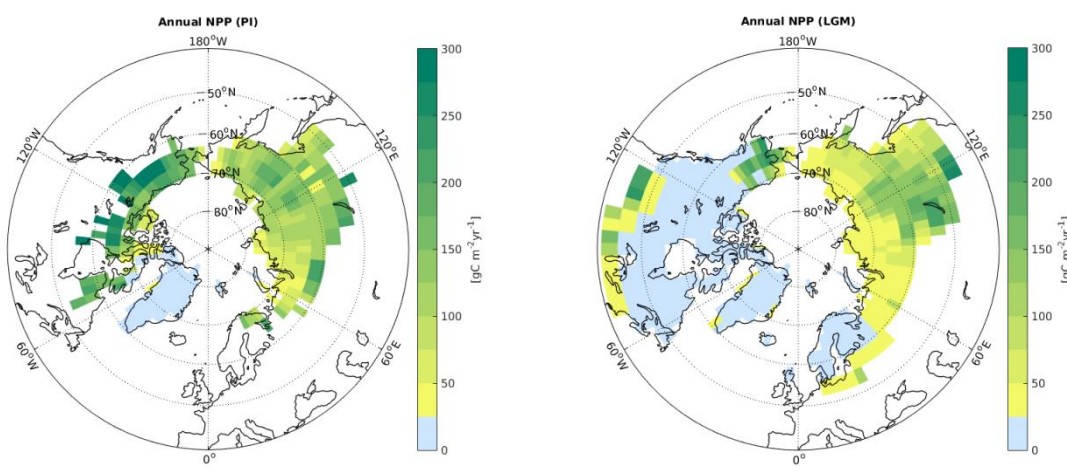

**Figure 4: Vegetation productivity (NPP, summed over all vegetation types) in near-surface permafrost regions for PI and LGM simulated by JSBACH. Light bluish areas show prescribed ice sheet coverage. Data shown represent hundred year time averages.**

The low bias in vegetation productivity proved especially critical for simulating glacial permafrost SOC storage (see next section), as low levels of glacial temperatures, precipitation and $CO_2$ concentration were strong limiting factors for glacial NPP and eventual SOC storage. The harsh simulated LGM climate pushes many permafrost grid cells close to bio-climatic vegetation limits. Therefore a small temperature bias can result in a strong underestimation of SOC storage in permafrost grounds, pointing to the importance of calibrating simulated LGM climate for modelling permafrost SOC build-up.

### 4.1.4    SOC storages at LGM and PI

Simulated SOC storage in the active layer ($SOC_{AL}$) shows pronounced regional to continental-scale differences for PI and LGM (Figure 5, upper panels). Pre-industrial SOC storage in North America is much higher than in Eurasia as a consequence of differences in simulated vegetation productivity in JSBACH. Given low glacial vegetation productivity, many grid cells reveal low LGM soil carbon storages below 5 kgC m$^{-2}$. The implementation of our newly developed scheme to account for SOC accumulation in perennially frozen ground has led to pronounced increases in total SOC storage in many regions compared to a reference run without accounting for permafrost carbon. The largest increases are inferred for northern

grid cells with rather shallow active layer that have >15 kgC m⁻² in permafrost (SOC$_{PF}$, Figure 5, lower panels). Under PI climate conditions, the largest permafrost SOC carbon accumulation is inferred for northernmost grid cells in Siberia where active layers are shallow, but where surface air temperatures are still high enough to support litter input to the soils. In contrast, under LGM climate conditions we infer a southward shift of maximum permafrost SOC as vegetation productivity in the northernmost grid cells decreases strongly.

Field data of mineral horizons in loamy permafrost-affected soils of Kolyma lowlands have organic carbon contents of 1-3%, with possible peaks up to 7%, likely due to cryoturbation (Mergelov and Targulian, 2011). These soils contain 7-25 kgC m⁻², which is within the range of our simulation results.

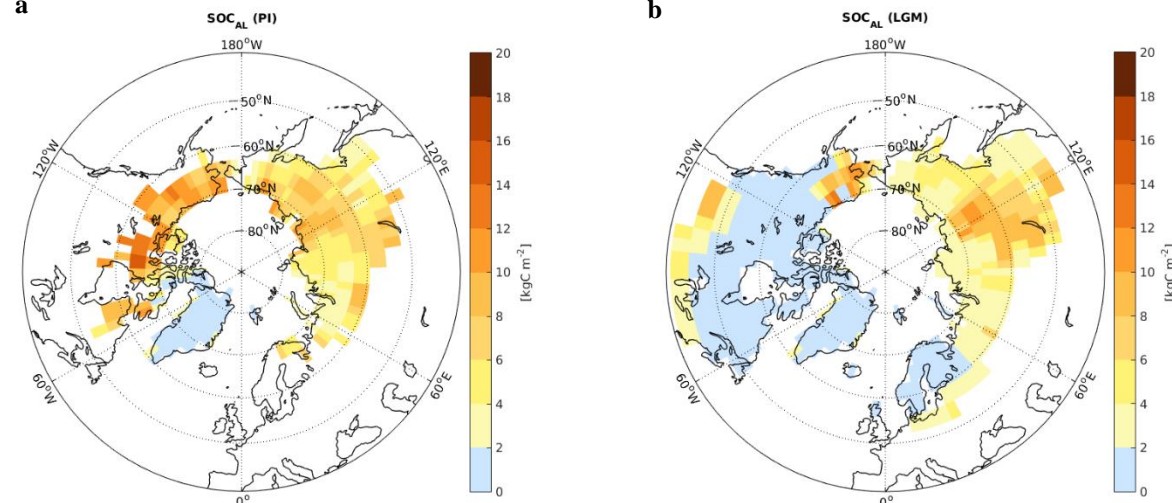

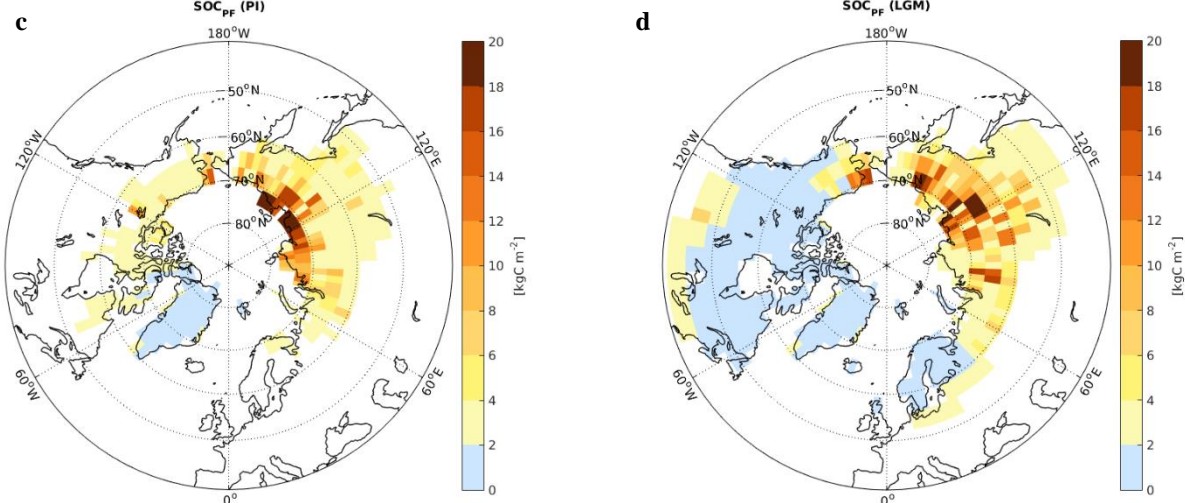

**Figure 5: Seasonally thawed SOC in the active layer (SOC_AL - panels a,b) and perennially frozen SOC in near-surface permafrost (SOC_PF - panels c,d) at PI and LGM simulated by JSBACH. Light bluish areas show prescribed ice sheet coverage. Data shown represent hundred year time averages.**

In contrast to observational data-sets, such as NCSCD, we represent SOM quantity *and* its degree of decomposition by our simulation approach (see section 7.7.1). Weighted over the permafrost domain, we model a SOC composition in the seasonally thawed soil surface of roughly equal shares of the slow (H) pool (~45-50%) and intermediate (N) non-soluble pool (~40%), with the remaining SOM stored in the fast pools. Simulated SOM stored in deeper perennially frozen ground reveals the imprint of depth-fractionation with a higher share of the slow pool (~60%). Southerly grid cells with deep active layers show largest slow pool fractions, which can reach 100% if seasonal thaw is pronounced. We therefore do not only capture accumulation of carbon in permafrost soils, but also model the depth dependent distributions of potential SOM lability, which in turn determines the timescale of C release upon thaw.

JSBACH simulates a total SOC storage in near-surface permafrost soils of 168 PgC under PI climate conditions, of which about a third is stored in permafrost (see also Figure 7 and

Table 2). When considering the full area of simulated permafrost ground (i.e. also considering grid cells with active layers below 3 meters depth), a total of 194 PgC is stored in permafrost soils. This is significantly lower than a recent empirically based reconstruction of LGM SOC stocks which includes ca. 800 Pg C in mineral soils, but for a larger permafrost area and under assuming that lower glacial $CO_2$ levels did not strongly reduce vegetation productivity (Lindgren et al., 2018). Given a multitude of factors which impact simulated SOC storage, current process-based permafrost-carbon models underline that uncertainty in simulated present-day permafrost carbon stocks is very large (McGuire et al., 2016) and suggest that our estimate falls in the lower range of model results.

Data-based estimates of pan-Arctic SOC storage in permafrost regions suggest a total of 1042 PgC (NCSCDv2.2(Hugelius et al., 2013). This amount also comprises SOC contributions from soils within the permafrost domain that are not underlain by

permafrost. When focusing on continuous and discontinuous permafrost and constraining the NCSCD data to grid cells with permafrost coverages larger than 50%, an estimated 575 PgC is stored in northern Gelisols. As we do not model organic soils (>40 cm surface peat; histels), we have re-calculated the Gelisol SOC estimate for model-data comparison purposes by assuming that histels have the same SOC concentrations as mineral cryoturbated soils (i.e. turbels) and infer a total of 547 Pg SOC storage.

As discussed above, part of the low bias in simulated SOC storage can be explained by low litter input into the soils due to an underestimate in vegetation productivity. Simulated vegetation productivity is especially low under the glacial climate as a consequence of low levels of glacial temperatures, precipitation and $CO_2$ concentration which prove a strong limiting factor for glacial NPP and eventual SOC storage. It is worth noting that the reconstruction by Lindgren et al. (2018) did not consider a potentially lower SOC stock due to $CO_2$ limitation at LGM times. In our model the harsh simulated LGM climate and low $CO_2$ levels pushes many permafrost grid cells closed to bio-climatic vegetation limits. Therefore a small temperature bias can result in a strong underestimate of SOC storage in permafrost grounds, pointing to the importance of calibrating simulated LGM climate for modelling permafrost SOC build-up.

Another contribution to differences in modelled and observed SOC storage is likely to come from SOM decomposition timescale assumptions. In section 4.3 we discuss an increase in simulated SOC accumulation due to increasing the residence time of the slow pool. These results underline that an accounting for processes of long-term SOM stabilization (e.g through the interaction with mineral compounds) would further increase simulated long-term SOC storages and should be considered an important aspect for further model development. A further aspect with regard to an improved representation of soil decomposition in permafrost will come from accounting for the full vertical soil temperature profile instead of tuning decomposition parameters to surface climatology only (see section 2.4).

Despite a slightly larger simulated permafrost extent under LGM conditions, total LGM SOC storage of 147 PgC is lower than PI storage due to a reduced vegetation productivity under harsh glacial climate conditions (see Figure 4). In contrast, using the land surface model ORDCHIDEE-MICT Zhu et al. (Zhu et al., 2016) infer much larger LGM SOC storage in permafrost regions of about 1220 PgC (without accounting for fast sedimentation in Yedoma regions). This large discrepancy results from of a factor of two lower LGM permafrost extent and much lower glacial vegetation productivity simulated by JSBACH compared to ORCHIDEE-MICT.

## 4.2     Deglacial climate and carbon dynamics

The dynamics of total SOC storage in permafrost are determined by the interplay of changes in permafrost extent and active layer thickness, as well as by changes in soil carbon net fluxes determined by litter input and losses mainly due to heterotrophic respiration. These factors are analysed in detail in the following sections.

### 4.2.1 Deglacial evolution of permafrost extent and SOC storage

Deglacial changes in permafrost extent are strongly shaped by the retreat of northern hemisphere ice sheets. Especially the decline in the Laurentide ice sheet has exposed large areas of soil in North America to cold air temperatures, which led to a build-up of permafrost in these regions. As a consequence, JSBACH simulates the maximum in permafrost extent not during the LGM with maximum ice-sheet coverage, but around 13 kyrs BP due to an increase in permafrost extent in North America (Figure 6). Consequently, deglacial warming results in a strong decline in total permafrost extent towards the beginning of the Holocene at 10 kyrs BP, while changes in permafrost extent over the Holocene period are less pronounced.

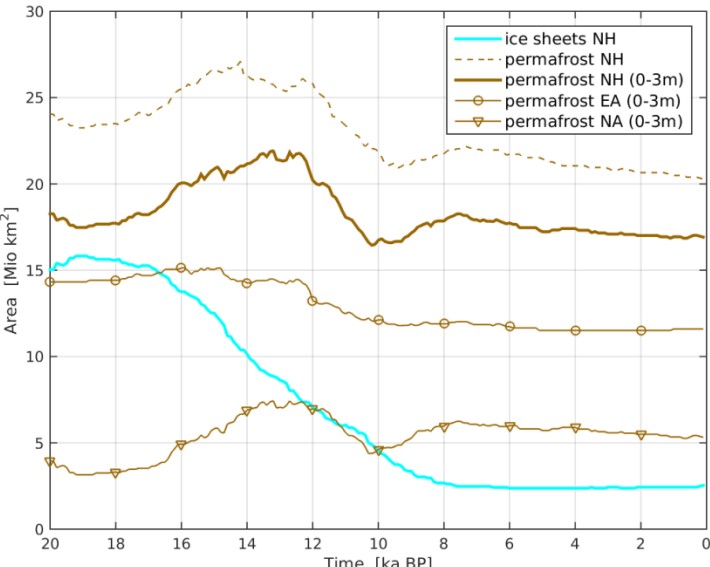

**Figure 6: Deglacial evolution of simulated permafrost extent and prescribed ice sheet area from LGM to PI. Permafrost extent (brown lines) is shown separately for total (dashed line) and near-surface (solid line) northern hemisphere permafrost, subdivided into North America (triangeled line), and Eurasia (circled line). Data shown represent hundred year time averages.**

Deglacial climate dynamics have also affected permafrost carbon storage by increases in vegetation productivity through higher $CO_2$ levels, and a prolonged and warmer growing season (thus increasing litter input to the soils). In parallel, soil respiration rates have increased by soil warming and carbon was transferred from perennially frozen to seasonally thawed soils through active layer deepening. Figure 7 shows the deglacial evolution of total SOC stored in near-surface permafrost along with individual contributions from active layer ($SOC_{AL}$) and permafrost ($SOC_{PF}$) organic carbon.

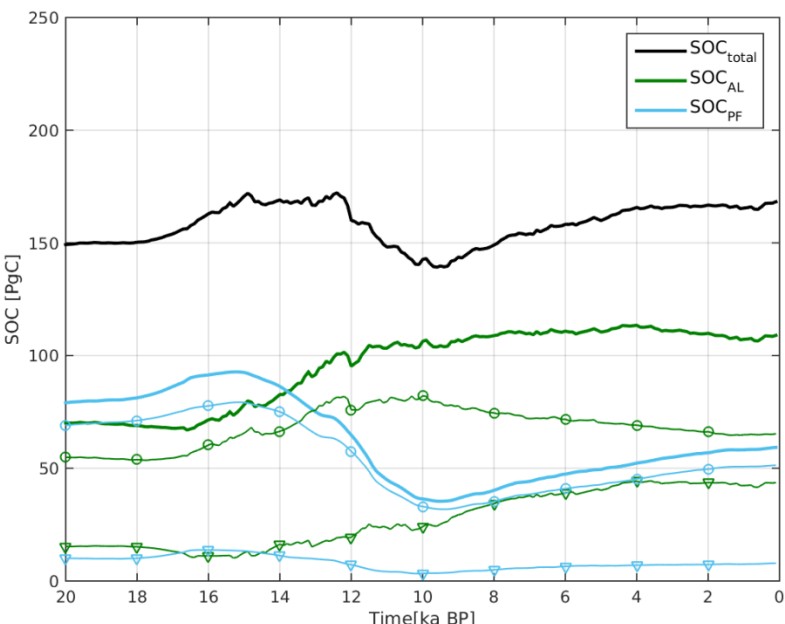

**Figure 7: Deglacial evolution of seasonally thawed and perennially frozen SOC in near-surface permafrost from LGM to PI. Total SOC (black line) is composed of seasonally thawed SOC (green lines) and perennially frozen SOC (blue lines). Contributions from North America (triangeled lines) and Eurasia (circled lines) are shown separately. Data shown represent hundred year time averages and were summed over near-surface permafrost grid cells (accounting for temporally evolving permafrost extents).**

Pronounced changes in permafrost SOC become evident after 17 kyrs BP, showing alternating phases of total permafrost carbon release and storage. Over the full deglacial period from the LGM to PI, we model a net accumulation of 29 Pg C from SOC in near-surface permafrost – in line with a recent empirically-based reconstruction of higher SOC accumulation in mineral soils of the permafrost domain under present day conditions as compared to the LGM (Lindgren et al. 2018). Changes in the pools of seasonally thawed and perennially frozen carbon are more pronounced. Total organic carbon storage in the active layer ($SOC_{AL}$) is gradually increasing from the Glacial towards the Holocene, while perennially-frozen organic carbon ($SOC_{PF}$) decreases strongly under deglacial warming towards the Holocene, and then increases slightly towards reaching PI conditions.

With regard to continental-scale deglacial SOC evolution, the temporal dynamics of permafrost coverage and SOC storage turn out to be only weakly linked (see Figure A10). Under climate warming, the direct loss of permafrost extent lowers the total amount of SOC stored in active layer and permafrost grounds. At the same time, this SOC loss is compensated by an increased litter input to permafrost soils due to higher vegetation productivity. For instance, the strong warming at 13 to 10 kyrs BP results in a pronounced reduction in North American and Eurasian permafrost extents (Figure A10) and increased soil respiration, but related SOC losses are more than outweighed by concurrent increases in NPP which finally result in

increased total $SOC_{AL}$ accumulation (Figure A10). A further consequence of climate warming is the deepening of active layers, which decreases the stock of perennially frozen SOC ($SOC_{PF}$) in favour of active layer SOC as carbon is transferred from frozen to thawed pools. In the case of permafrost area gains as a consequence of ice sheet retreat, e.g. between 17 and 13 kyrs BP in North America (Figure A10), the total area of permafrost SOC accumulation is increasing. Yet the climate has

to warm sufficiently to favour intense vegetation productivity in newly established permafrost grid cells which explains the time lag in $SOC_{AL}$ increases. Figure A10 underlines that the evolution of active layer SOC is rather tracked by changes in NPP in permafrost regions than by changes in permafrost extent. In Eurasia, a long term shallowing trend in simulated active layer depths after 10 kyrs BP results in a NPP decline towards reaching the PI climate in our model. The sensitivity of simulated NPP to active layer depth could explain how climatology biases affect permafrost region vegetation productivity

via soil moisture coupling and therefore negatively affect permafrost SOC accumulation (especially in cold-biased regions of Siberia).

In contrast to Crichton et al. (2016), we infer a slight increase in permafrost SOC storage between 17.5 and 15 kyrs BP instead of a large release of thawed permafrost carbon. Part of the discrepancy in simulated permafrost carbon dynamics can

be explained by modelling rather different trajectories of deglacial permafrost extents. We also do not capture abrupt permafrost carbon release as suggested by Köhler et al. (2014) during the onset of the Bølling-Allerød. The authors hypothesize that the source of old [14]C depleted carbon was eventually affected by large contributions from flooding of the Siberian continental shelf – a potential carbon source region which we do not capture by our modelling approach. With a focus on global $CH_4$ levels (which we do not consider in this study), $CH_4$ release from newly forming thermokarst lakes was

postulated to have strongly affected global $CH_4$ levels at the Pleistocene-Holocene transition (Walter et al., 2007).
Using an Earth System Model of intermediate complexity, Ganopolski and Brovkin (2017) have recently analysed the contribution of terrestrial and ocean processes to the glacial-interglacial $CO_2$ cycle. The important role of permafrost carbon in these simulations is that it lowers the increase of total deglacial terrestrial carbon, which is caused by $CO_2$ fertilization and re-establishment of boreal forests. This study also found a minimum in permafrost carbon storage at the beginning of the

Holocene, in line with our results.

### 4.3     Sensitivity runs

We have tested the robustness of our simulations with regard to model parameter choices which affect active layer depths, vertical SOC profiles, and decomposition timescale. Further, we have run an additional experiment in which we have doubled litter input to the YASSO soil carbon module to compensate for low biases in simulated vegetation productivity.

Table 2 shows how the individual sensitivity experiments affect simulated SOC storages under LGM and PI conditions.

| Experiment | $SOC_{tot}$ [PgC] | | $SOC_{AL}/SOC_{PF}$ | | $SOC_{tot}/PFA$ [kgC/m$^2$] | |
|---|---|---|---|---|---|---|
| | LGM | PI | LGM | PI | LGM | PI |
| **L2P** | 147 | 168 | 0.94 | 1.85 | 8.0 | 9.9 |
| **L2P_ALD** | 149 | 271 | 0.42 | 0.64 | 7.7 | 14.1 |
| **L2P_VMR** | 152 | 172 | 0.83 | 1.65 | 8.4 | 10.2 |
| **L2P_HDT** | 194 | 205 | 0.87 | 1.77 | 10.6 | 12.1 |
| **L2P_LIT** | 288 | 320 | 0.89 | 1.83 | 15.8 | 19.0 |
| **L2P_CTR** | 71 | 111 | - | - | 3.9 | 6.5 |

**Table 2: Simulated soil organic carbon storage under LGM and PI conditions in near-surface permafrost soils for the standard parameter setting (L2P), reduced thermal conductivity of the organic layer (L2P_ALD), increased vertical soil mixing (L2P_VMR), increased slow pool lifetime (L2P_HDT), doubled litter input (L2P_LIT), and a control run without transfer of SOC to permafrost (L2P_CTR). Storages are expressed as totals, as the ratio between active layer and permafrost carbon, and normalized by the near-surface permafrost area (PFA).**

When decreasing the organic surface layer conductivity by a factor of two (experiment L2P_ALD), we model a slight increase in permafrost extent (by 13.6% for PI, by 6.0% for LGM). Yet, the dominant effect on simulated permafrost soils is manifested in shallower active layer depths (see Figure A9), thus shifting the weight between active layer and permafrost carbon towards a larger carbon store in permafrost layers (Table 2). This increase in the fraction of permafrost carbon favours SOC accumulation by reducing heteorotrophic respiration losses. Under LGM conditions, this SOC gain is compensated by simulating very shallow active layer depths in many grid cells which result in lower vegetation productivity in L2P_ALD compared to L2P as a consequence of modified soil moisture and soil water availability. Therefore, simulated total LGM storage is comparable in both experiments (table 2). In contrast, $SOC_{tot}$ under PI conditions is about 60% larger (amounting to 271 PgC) in L2P_ALD compared to our standard parameter setting (L2P).

We have further investigated, how a doubling of the cryoturbation rate in the process-based model of SOC accumulation is affecting vertical SOC distributions, and therefore simulated SOC transport between active layer and permafrost carbon pools. We infer a slight increase in the fraction of permafrost SOC as a consequence of the faster SOM transport through the active layer, but the overall effect on simulated SOC storages is rather small (table 2). Of larger impact is uncertainty in the assumed decomposition time of the slow pool. After increasing the slow pool turnover time in YASSO from 625 years (L2P) to 1000 years (L2P_HDT), we infer an increase in total SOC storage of 31.8% (LGM) and 22.0% (PI). An increase in simulated SOC storage would also result if YASSO soil decomposition parameters were scaled by soil instead of surface air temperatures (see discussion in section 2.4). The extent to which SOC storage will increase is uncertain and an improved description of temperature sensitivity of decomposition is subject to current JSBACH model development. Finally, we have run an additional experiment with doubling of the soil litter input to YASSO to compensate for our inferred low bias in vegetation productivity. Simulated SOC stores in L2P_LIT almost double and amount to 288 PgC (LGM) and 320 PgC (PI), reducing the mismatch to observational data (see discussion in section 4.1.4).

## 5 Conclusions

Using a new land surface model offline version of JSBACH we have simulated long-term permafrost carbon dynamics from the Last Glacial Maximum (LGM) to Pre-industrial (PI) climate, driven by climate forcing fields generated from MPI-ESM (version 1.2 in T31GR30 resolution). Focusing on continuous and discontinuous permafrost extent, we simulate a near-surface permafrost extent (i.e. permafrost in the upper three meters of the soil) of 16.9 Mio km$^2$, which is close to observational estimates. Simulated near-surface permafrost extent under glacial conditions during the LGM shows a pronouncedly different spatial pattern, with slightly increased total area coverage of 18.3 Mio km$^2$. Empirical reconstructions of LGM permafrost suggest a larger areal extent, mainly due to an underestimate of JSBACH in simulated LGM permafrost in Europe, western and central Asia compared to the reconstructions. Despite comparatively small simulated changes in total permafrost extent between LGM and PI, our simulations show broad-scale shifts in permafrost coverage, with permafrost disappearance in southerly regions, and permafrost aggregation in formerly ice-covered grid cells in North America during deglacial warming.

The implementation of our newly developed model to calculate soil organic carbon (SOC) accumulation in JSBACH in perennially frozen ground has strongly increased total simulated SOC storage at high latitudes: We model a LGM SOC storage of 72 PgC in seasonally thawed soil layers comprising all grid cells with permafrost in the upper three meters. When additionally accounting for SOC accumulation in perennially frozen soil layers, which prevents permafrost organic matter from decomposition, we infer a total SOC storage of 147 PgC – doubling the amount of simulated LGM SOC in a control experiment with identical permafrost physics but without modelling carbon transport to permafrost layers.

Simulated deglacial warming triggers pronounced changes in regional permafrost extent and active layer depths. In parallel, litter input into the soils increases through higher vegetation productivity, while soil respiration increases due to warming temperatures. As a consequence of combined deglacial changes in physical and biogeochemical driving factors we infer an increase in total permafrost SOC storage towards the Holocene (168 PgC at PI), with largest changes seen in the individual contributions of permafrost and active layer carbon. Our modelled PI SOC storage is low compared to observations of total carbon stored in soils of the permafrost region (~1300 PgC) as we do neither model high soil carbon accumulation in organic soils, nor in soils with little or no  permafrost or in deep deposits within the permafrost region. When focusing on near-surface permafrost sites of continuous and discontinuous occurrences (describing a gelisol coverage larger than 50%), observations suggest a total of 575 Pg of permafrost soil carbon. We inferred an improved agreement of simulated permafrost SOC storage with observational data when compensating low vegetation productivity in our coarse resolution model version (MPI-ESM1.2T31) by doubling soil litter input in JSBACH, which leads to a storage of 320 Pg of SOC under PI climate conditions. Additional model experiments with JSBACH revealed the sensitivity of simulated SOC storage in permafrost regions to slow pool decomposition timescale and to active layer depths. A larger storage of pre-industrial SOC was inferred when increasing the slow pool turnover time (205 PgC), and when increasing the insulation of the organic

surface layer (271 PgC). Not capturing processes of long-term SOM stabilization in our model (e.g. through the formation of organo-mineral associations) can further explain a part of model-data differences.

Rather than a steady increase in carbon release from the LGM to PI as a consequence of deglacial permafrost degradation, our results show alternating phases of permafrost carbon release and accumulation, which illustrates the highly dynamic

nature of this part of the global soil carbon pool. The temporal evolution of active layer SOC proved to be strongly linked to changes in NPP in permafrost regions, rather than to changes in permafrost extent which can be explained by pronounced time lags between establishment of new permafrost after ice sheet retreat and onset of intense vegetation productivity. Our simulations show a long-term shallowing trend of active layer depths towards reaching the PI climate which results in a sustained but slow transfer of active layer SOC to perennially frozen pools after 10 kyrs BP.

Over the full deglacial period from the LGM to the PI climate, we model a net accumulation of 21 PgC in near-surface permafrost soils (i.e. an increase by 14% above LGM SOC). The full extent to which carbon accumulation and release as a consequence of deglacial permafrost degradation has likely affected past variations in atmospheric glacial-interglacial greenhouse gas levels depends critically on the realism of simulated glacial vegetation productivity and permafrost thermal state which both are subject for future model improvements.

## 6    Acknowledgements

Primary data and scripts used in the analysis and other supplementary information that may be useful in reproducing the author's work are archived by the Max Planck Institute for Meteorology and can be obtained by contacting publications@mpimet.mpg.de. T.S.v.D. acknowledges support from BMBF projects CARBOPERM (grant 03G0836C),

KOPF (grant 03F0764C), and PERMARISK (grant 01LN1709A). G.H. acknowledges the EU-JPI COUP consortium and a Marie Curie Skłodowska and Swedish Research Council International Career Grant (INCA; no. 330-2014-6417). We thank N.Carvalhais for providing a compilation of soil depth data and A. Ganopolski for the provision of glacial-interglaical simulation data which we used to generate forcing anomalies. Philipp de Vrese has reviewed the manuscript and contributed helpful feedback for improving the paper. Many thanks also to Veronika Gayler, Rainer Schnuur, and Thomas Raddatz for

discussing JSBACH model aspects.

## 7    Appendix

### 7.1    Applied climate forcing fields for a transient deglacial simulation with JSBACH

We have performed experiments with a standalone configuration of the MPI-ESM land surface model JSBACH, driven with climate forcings derived from coupled climate time-slice model experiments in coarse T31 resolution performed under

preindustrial and glacial conditions with MPI-ESM1.2 (as described in Mauritsen et al. (2018), with differences to the base

version described in Mikolajewicz et al. (2018)). As the availability of MPI-ESM1.2 experiments is limited to these two time slices, we follow an anomaly approach for modelling deglacial climate dynamics and used climatic fields from a transient glacial cycle experiment with the intermediate-complexity ESM CLIMBER2 (Ganopolski et al., 2010).

For the study presented here, we use the MPI-ESM1.2 preindustrial climate experiment (described in section 7.2) as the basis of the climate forcings. Climate forcings for earlier times were derived by applying monthly anomalies to the preindustrial climate, with absolute anomalies used for surface air temperature fields and relative anomalies for precipitation, humidity, radiation and wind speed. The anomaly applied to the MPI-ESM PI climate is derived as a linear interpolation between MPI-ESM LGM and CLIMBER2 anomalies, depending on the distance of CLIMBER2 global mean temperature to the LGM state. The weight of the MPI-ESM anomaly in this interpolation is shown in Figure A1.

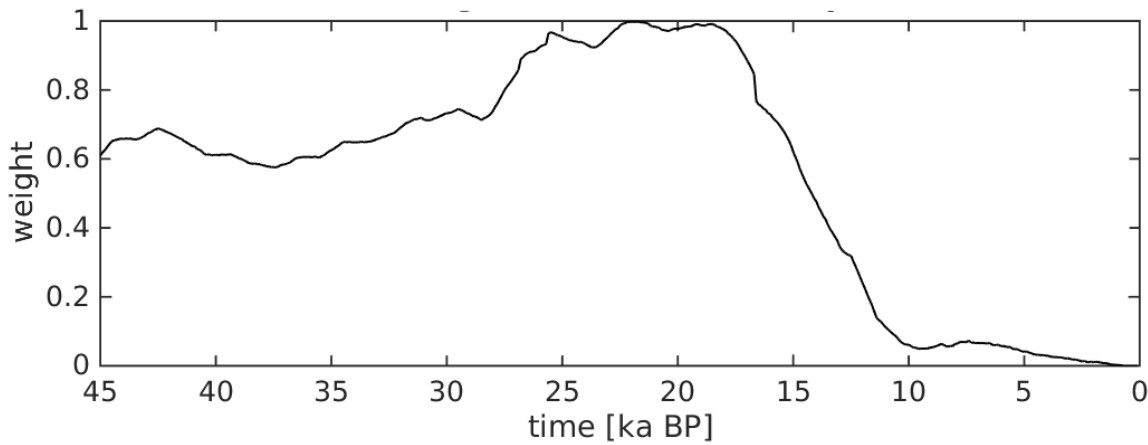

**Figure A1: Transient weighting of MPI-ESM anomaly (LGM-PI). At LGM, the anomaly is fully described by MPI-ESM. CLIMBER2 anomaly weight is 1 – (MPI-ESM anomaly).**

This procedure of determining the climatic anomalies ensures that near-LGM conditions are derived from the high-resolution MPI-ESM climatic fields, while climate during the deglaciation and the Holocene is derived from the lower resolution but spatio-temporally consistent CLIMBER2 fields.

## 7.2   PI climate

To derive the climate forcings for the standalone JSBACH model we performed an experiment with MPI-ESM in version 1.2 in resolution T31GR30 (MPI-ESM1.2T31, corresponding to ~ 400 km x 300 km at 45°N) using a half-hourly time-step. We add a brief description of the northern high latitude climate and differences to the CMIP5 version of the model. In CMIP5

the standard (LR) resolution was T63GR15, and climatic fields for comparison were interpolated to T31GR30 here. The CMIP5 preindustrial climate was described in (Giorgetta Marco et al., 2013).

In the version we are using here, the global mean temperature is 286.78 K, nearly identical to the global mean temperature of 286.66 K of the CMIP5 model in T63 resolution. However, the spatial distribution of temperature is modified, as shown in Figure A2. Annual mean temperatures over northern North America are warmer than the CMIP5 reference, while temperatures over Eurasia are cooler. This spatial pattern is also affecting simulated pre-industrial vegetation productivity (Figure A4) which shows higher GPP in North America and lower GPP in Eurasia compared to observational evidence based of up-scaled flux tower measurements (Jung et al., 2011).

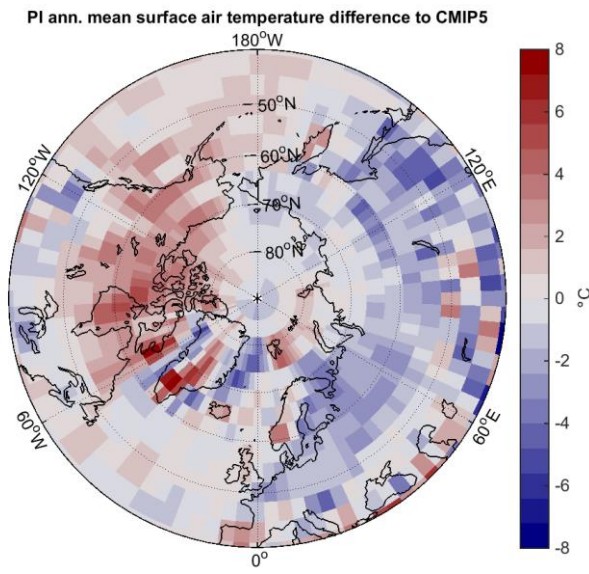

**Figure A2: Difference in simulated annual mean surface air temperature by MPI-ESM1.2T31_PI compared to CMIP5 PI experiment.**

As shown in Figure A3, PI winter (DJF) temperatures reach -35° C over Greenland and northern Siberia. Southern Siberia and North America are warmer, though temperatures warmer than -10°C are limited to latitudes south of 50° N. Summer (JJA) conditions are substantially warmer, with temperatures below freezing only over Greenland, and most NH high latitude areas having summer temperatures of the order of 10° C. Annual precipitation, shown in Fig.A3 is less than 750 mm yr$^{-1}$ over most of the high latitude regions, and less than 500 mm yr$^{-1}$ north of 65° N.

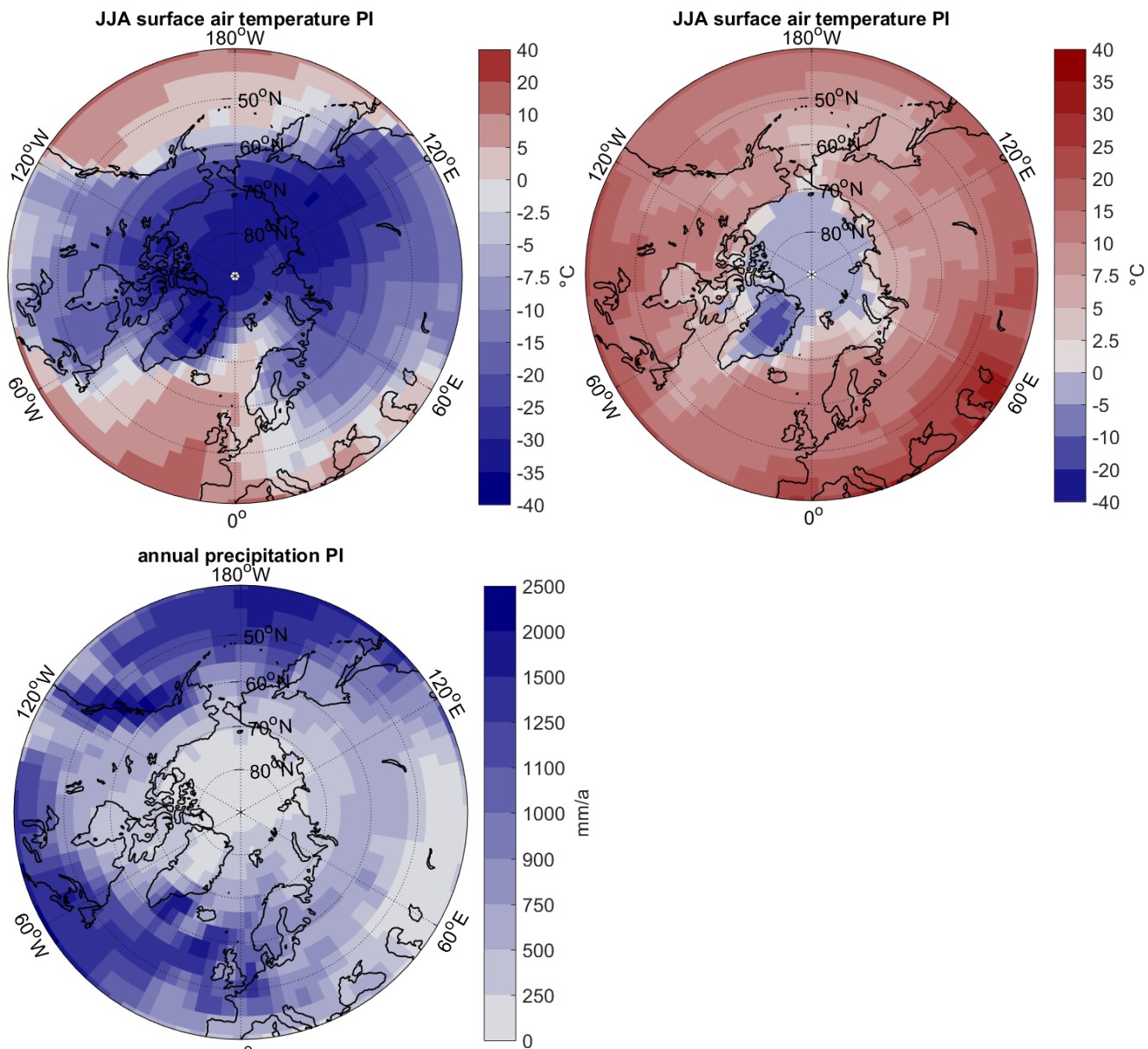

**Figure A3: PI winter surface air temperatures (DJF), summer surface air temperature (JJA) and annual precipitation simulated by MPI-ESM1.2T31_PI.**

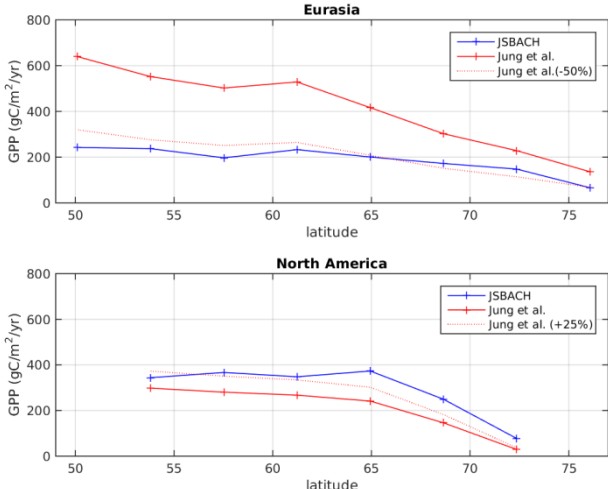

**Figure A4: Simulated and observed high latitude GPP for Eurasia and North America. Model results are for pre-industrial conditions. Observations are from Jung et al. (2011) and scaled to 85% to account for lower pre-industrial GPP (Ciais et al., 2013). Regional zonal averaging was only performed for grid cells containing near-surface permafrost.**

## 7.3   LGM climate

The LGM climate experiment is set up following the PMIP3 protocol (Braconnot et al., 2011), with LGM land-sea and ice sheet masks, as well as greenhouse gases and orbit modified to LGM conditions. The global mean surface air temperature is 282.94 K, 3.84 K colder than for PI (which is at the lower end of PMIP3 model results, (Schmidt et al., 2014)). The

10   differences in annual mean temperatures, shown in Figure A5, are largest over the Laurentide ice sheet, where cooling is up to 30° C. In Siberia the cooling is about 8 ° C in the annual mean.

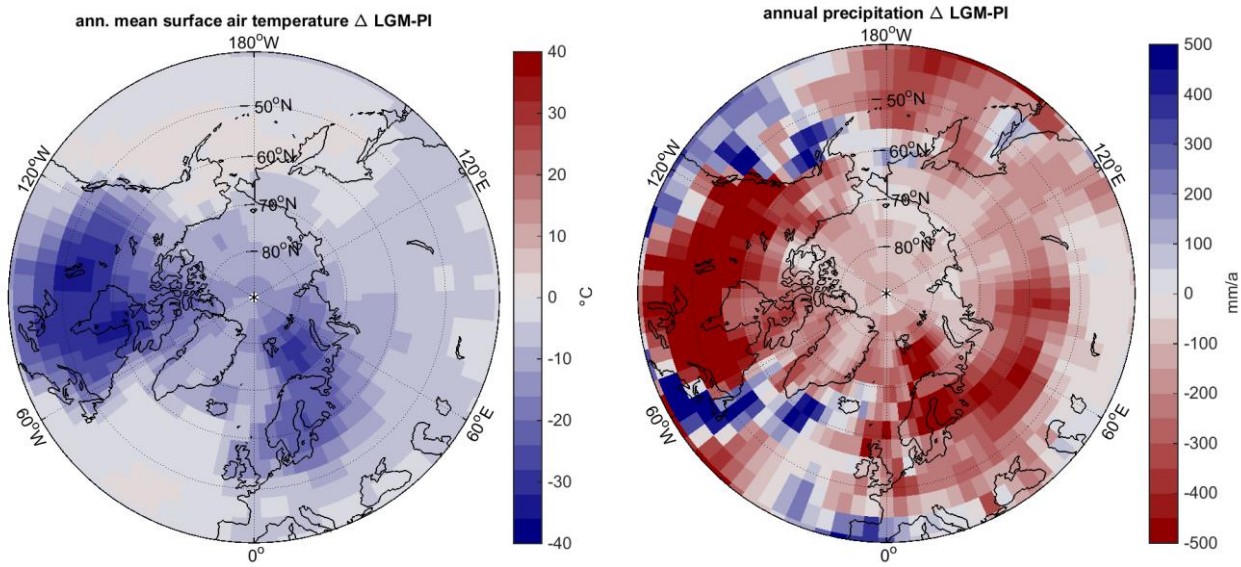

**Figure A5: Difference in (a) annual mean surface air temperatures and annual mean precipitation between LGM and PI as simulated by MPI-ESM1.2T31**

Precipitation changes, shown in Figure A7, show a general drying trend, with pronounced differences east of the Fennoscandian ice sheet where precipitation reduces by 200-400 mm/a.

## 7.4    Boundary conditions

### 7.4.1    Ice sheet extent

Ice sheets extent is prescribed from a transient glacial cycle experiment performed with CLIMBER2 and the ice sheet model SICOPOLIS (Ganopolski et al., 2010). As SICOPOLIS ice sheet extent for LGM is slightly larger than the ice sheet extent used in the MPI-ESM LGM experiment, we limit ice sheet extent to the MPI-ESM LGM ice sheet mask, shown in Figure A6.

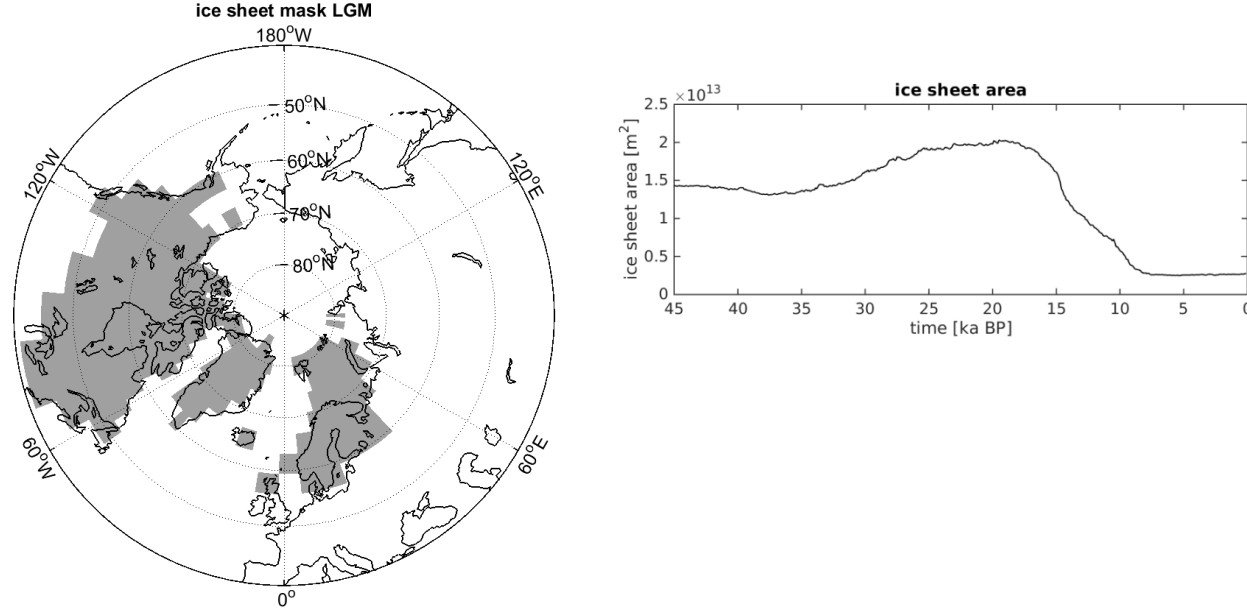

**Figure A6: (a) Ice sheet extent at LGM prescribed in MPI-ESM1.2T31_LGM, and (b) evolution of total NH ice sheet area from 45 kyrs BP to PI simulated by CLIMBER-2.**

5    The transient total ice sheet area, shown in Figure A6, is maximal at 20 ka BP (20.3 million square kilometers), while the PI size is 2.76 million square kilometers. As the JSBACH ice sheet and land-sea masks cannot be varied during run time, we keep these fixed at PI extent. However, the effect of ice sheets on vegetation and soil carbon is represented by removing precipitation in ice sheet locations, thereby preventing the development of vegetation and soil carbon accumulation.

10   **7.4.2    CO$_2$ concentrations**

Atmospheric CO$_2$ contents are prescribed following CLIMBER-2 glacial cycle experiments (Ganopolski et al., 2010) and are shown in Figure A7.

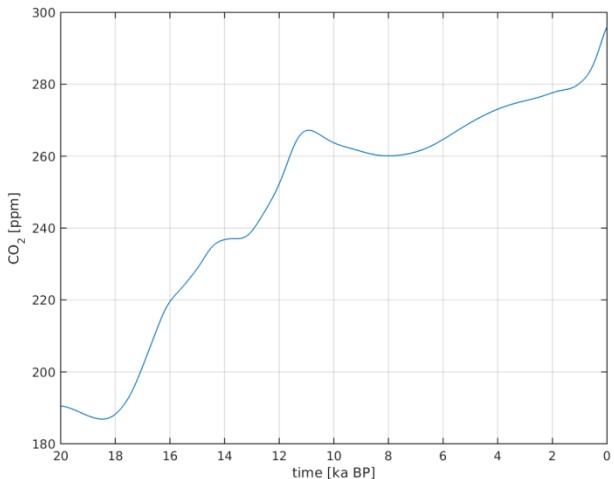

**Figure A7: Atmospheric CO₂ equivalent concentration from 20 ka BP to PI prescribed in JSBACH. Data were obtained from a CLIMBER-2 glacial-cycle simulation (Ganopolski et al., 2010).**

### 7.4.3 Soil depths

We prescribed stationary soil depths in JSBACH based on a global soil map compiled by Carvalhais et al. (2014, Figure A8).

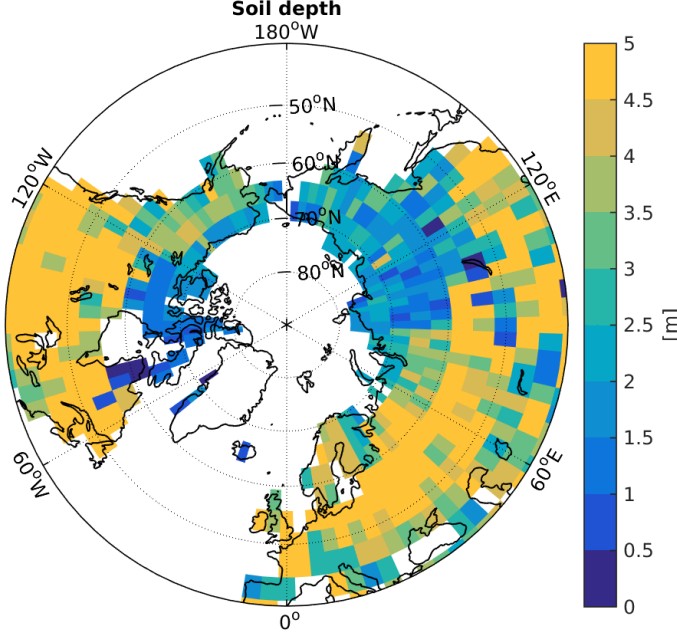

**Figure A8: Prescribed soil depths after Carvalhais et al. (2014).**

Bluish grid cells indicate regions in which LGM permafrost SOC accumulation in the upper three meter is constrained by soil depth. Using equation 2, we estimated the consequence of neglecting the limitation in SOC build-up through soil depth. Assuming a lower boundary of three meters for all near-surface permafrost grid cells, our simulated LGM permafrost carbon pool would be 33% (or 26 PgC) larger. By using constant soil depths, we implicitly assume that soil accumulation and erosion rates in non ice-covered grid cells were in equilibrium over the model simulation time horizon. For ice-covered grid cells we assume a full removal of soil through the ice movement.

Depth to bedrock is a poorly constrained variable also for concurrent soil C stock estimates. Jackson et al. (2017) showed that applying different products of soil depth led to differences in global soil C stocks of up to 800 PgC in the upper 3 m, with most of the differences occurring in high-latitude soils. Future model developments should analyse whether alternative soil depth data products (e.g. Pelletier et al. (2016) or Hengl et al. (2017)) might better capture soil depths in permafrost regions, possibly supporting less shallow soils and therefore larger LGM SOC storage. During the transient deglacial warming phase, permafrost SOC build-up is prevented in our model when the simulated active layer depth falls below soil depth, which is the case for 17% of permafrost grid cells under PI climate conditions. These grid cells accumulate 41 PgC in the active layer, but they do not allow for additional SOC accumulation in permafrost.

### 7.5 Comparison of simulated ALDs with CALM observations

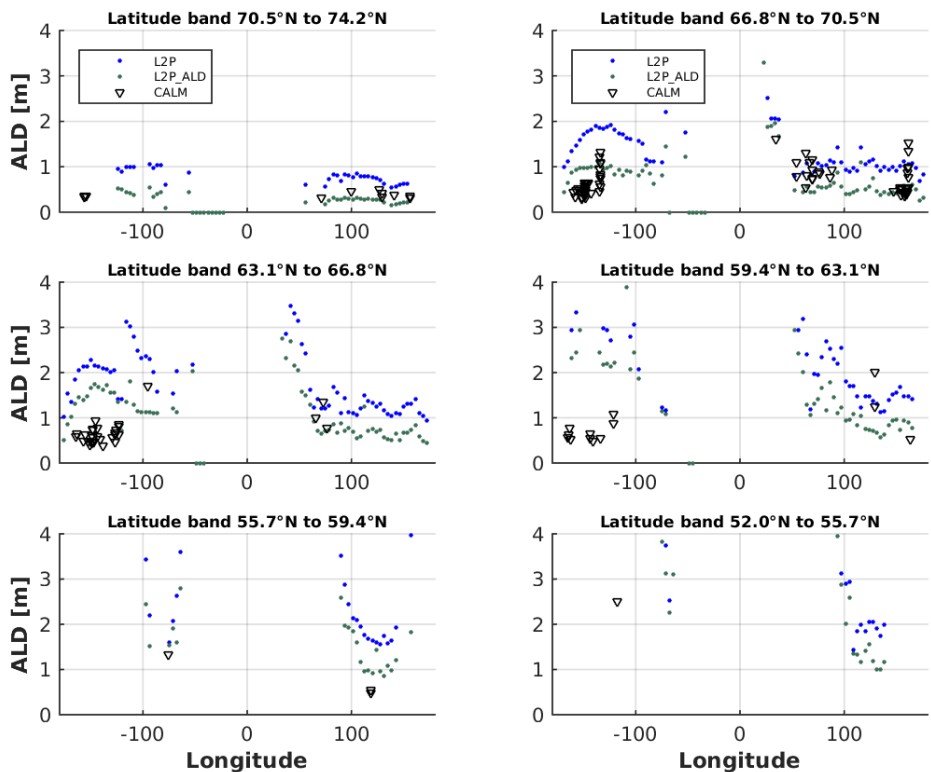

**Figure A9: Latitudinal dependency of active layer depths inferred from JSBACH simulations (dots) and CALM observations (triangles). Blue dots represent the standard model experiment (L2P), green dots the sensitivity run with increased thermal insulation of the organic surface layer (L2P_ALD).**

## 7.6    Driving factors of deglacial SOC dynamics

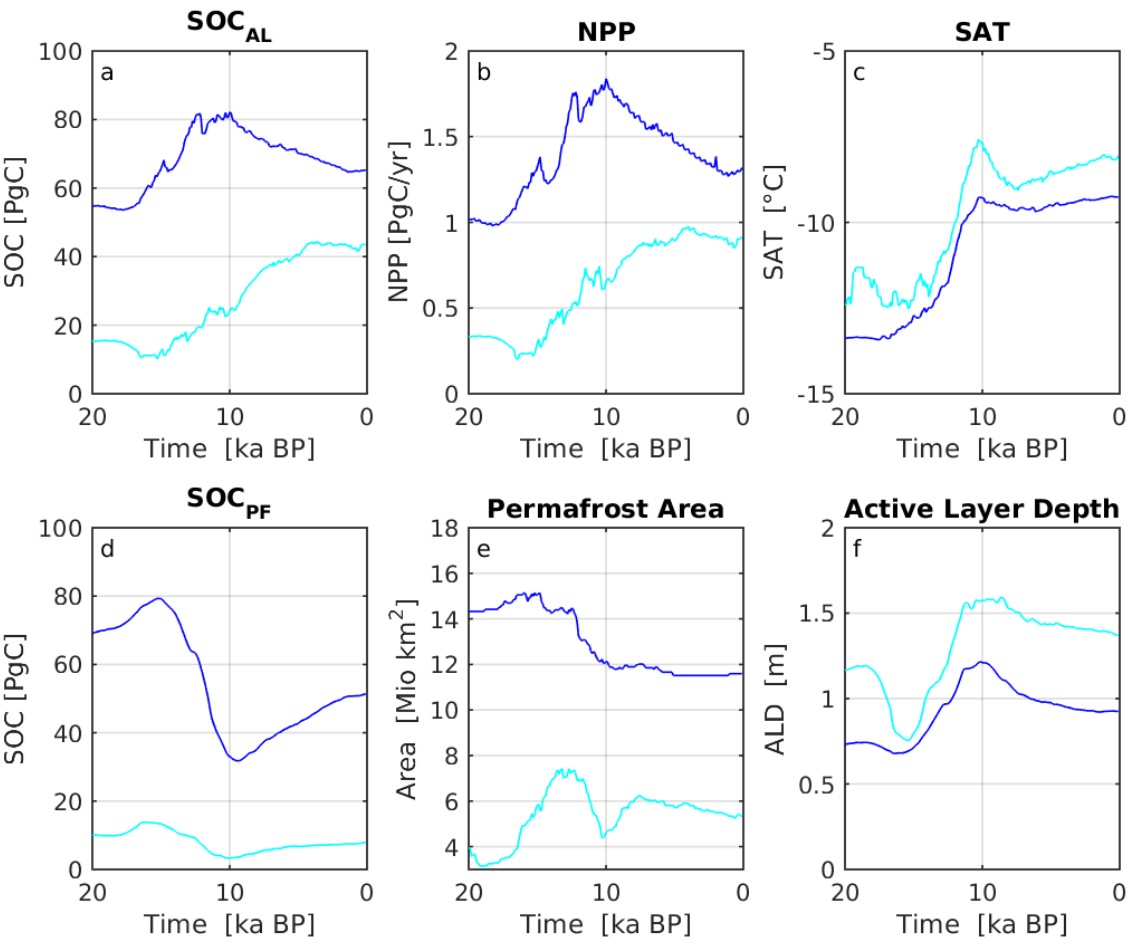

**Figure A10: Deglacial evolution of seasonally thawed (a) and perennially frozen SOC (d) in near-surface permafrost from LGM to PI. Panel (b) and (e) show deglacial evolution of NPP summed over near-surface permafrost grid cells, and permafrost extent. Panels (c) and (f) illustrate mean annual surface air temperature and active layer depth, which both were weighted over permafrost grid cells. Contribution from North America (light blue) and Eurasia (dark blue) are shown separately.**

## 7.7    Accounting for vertical SOC profiles in JSBACH

Permafrost soils reveal typical profiles of depth-declining soil organic carbon concentrations (SOCC) in the active layer (Harden et al., 2012). To capture this key characteristic, which strongly affects the amount of carbon transferred between seasonally thawed and perennially frozen carbon pools, we have developed a process-based model which allows the

calculation of vertical SOCC profiles dependent on factors such as active layer depths, lability of the organic matter, vertical soil mixing rates. In this section we describe the physics of this model and its implementation into JSBACH.

### 7.7.1 Modelling soil carbon profiles with a process-based model of SOM transport

Soil carbon build-up is modelled by assuming that the carbon flux balance in each soil layer is determined by litter input, by a transport of carbon through diffusion and advection within the soil column, and by loss of carbon through heterotrophic respiration within the active layer (Braakhekke et al., 2014):

$$\frac{\partial socc^i(z,t)}{\partial t} = F_{litter}^i(z,t) + D\frac{\partial^2 socc^i(z,t)}{\partial z^2} + \alpha\frac{\partial socc^i(z,t)}{\partial z} - \beta^i(z,t) * SOCC^i(z,t) \tag{A1}.$$

$SOCC^i(z,t)$ describes the soil carbon concentration (in kgC m$^{-3}$) for each soil carbon pool i (i=1..5, based on YASSO soil carbon separation). Litter input (in kgC m$^{-2}$) is assumed representative for grassland (being the dominant vegetation cover in permafrost regions) and is subdivided by equal shares into aboveground and belowground fluxes. While aboveground flux enters the uppermost soil layer only, belowground litter flux is restricted to the active layer and is described by a depth

profile according to Jackson et al. (1996):

$$F_{litter}^i(z,t) = F_{litter}^i(z_0,t) * \gamma^z \tag{A2},$$

with γ=0.943 chosen to represent the depth distribution of temperate grassland (z in cm). A re-scaling assures carbon closure,

especially for active layers less deep than theoretical root depths. $D$ describes a diffusivity parameter (in and mimics the strength of cryoturbation. As we focus in our study on mineral, cryoturbated soils, we do not separately describe soil carbon profiles inferred without vertical mixing, but we investigate the sensitivity of modifying the diffusivity parameter on our results (see section 4.3). We choose a default setting for $D$ of 10 cm$^2$ yr$^{-1}$ (Koven et al., 2009). As radiocarbon data provide only a weak constraint on cryoturbation strength, we acknowledge that this parameter is subject to large uncertainty and

investigate the robustness of our results for a doubled cryoturbation rate setting (see section 4.3). At depth levels below the active layer, the diffusivity parameter is set to zero. Under a stationary climate, only advection through sedimentation can transport SOM into the permafrost body. Based on SOC-age profiles from a loess-paleosol sequence in north-east Siberia (Zech et al., 2011) we choose a standard sedimentation rate α of 10 cm kyr$^{-1}$. Decomposition of soil organic matter is calculated according to YASSO (i.e. separating soil organic matter into four classes of differing litter lability and a slowly

decomposing and more stable component). Pool specific decomposition rates β$^i$ (z, t) (in yr$^{-1}$) are described dependent on temperature $T(z,t)$ at each soil level (in degrees Celsius):

$$\beta^i(z,t) = \beta_0^i * \exp(p_1 * T(z,t) + p_2 * T(z,t)^2) \hspace{2cm} \text{(A3),}$$

with reference decomposition rates $\beta_0^i$ at 0°C for each lability class according to YASSO standard parameters ($p_1 = 0.095$ °C$^{-1}$ and $p_2 = -0.0014$ °C$^{-2}$). For sub-zero temperatures decomposition is set to zero. Within the range of typical magnitudes in variations of soil temperatures and of soil moisture inferred from permafrost grid cells (without describing saturated soil conditions), we consider the impact of soil moisture on SOCC build-up being of secondary importance and describe SOC build-up depending on soil-temperature only. This assumption is in line with Bauer et al. (2008) and Exbrayat et al. (2013) who inferred a dominant control of simulated heterotrophic respiration exerted by soil temperature, while soil moisture effects revealed less important.

To determine SOCC$_{AL}$ values for a broad range of active layer depths, the model is driven by a range of annual temperature cycles of differing mean annual ground temperatures (MAGT) to generate active layer depths from 10 cm to 3 m. Soil carbon decomposition rates are calculated in each soil layer dependent on simulated soil temperatures and litter input is linearly scaled by MAGT (by covering typical litter inputs from JSBACH in permafrost grid cells). The scaling assumes 100 gC yr$^{-1}$ m$^{-2}$ at a MAGT of -16 °C, and 200 gC yr$^{-1}$ m$^{-2}$ at a MAGT of -10 °C, covering the active layer range between 50 cm and 150 cm shown in Figure A10. SOCC is finally calculated by solving Eq. (1) at each vertical grid level for each time-step.

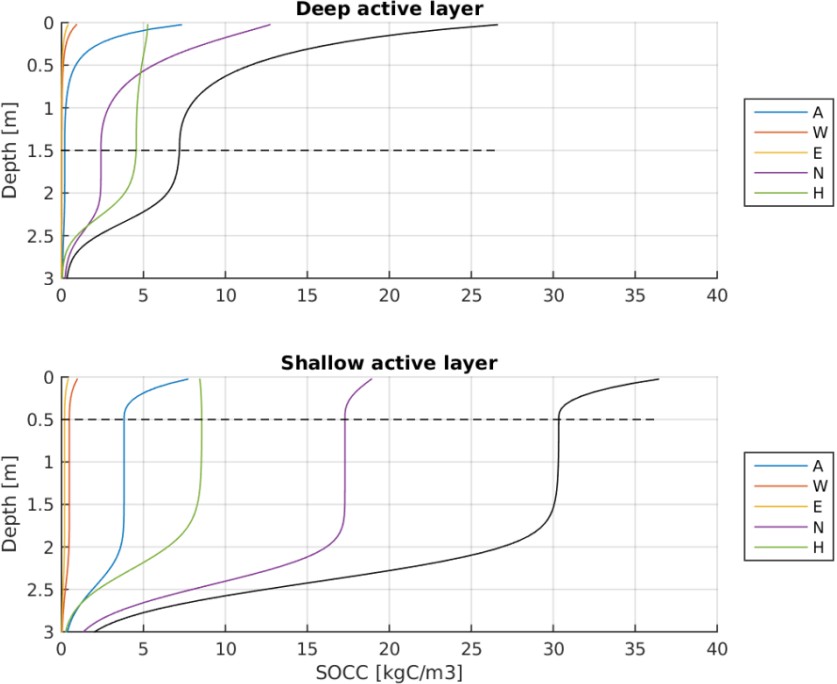

**Figure A11: Simulated SOCC profiles inferred from the process-based model of vertical SOM dynamics. The coloured curves represent different SOM lability classes according to YASSO, separated into fast (A,W,E), intermediate (N), and slowly**

**decomposing (H) compounds. The black line shows aggregated total SOC. The horizontal dashed lines indicate active layer depth, and determine $SOCC_{ALD}^{i}$ . Simulations were run for 20 kyrs using a cryoturbation rate of 10 cm$^2$ yr$^{-1}$, a sedimentation rate of 10 cm kyr$^{-1}$, decomposition timescale parameters set to YASSO standard values, and using a litter input of 100 gC yr$^{-1}$ m$^{-2}$ (upper panel) and 200 gC yr$^{-1}$ m$^{-2}$ respectively (lower panel).**

Once transported into permafrost layers, SOC is protected from microbial decomposition and establishes a depth-constant SOCC profile. The decline of SOCC in soil layers below about two meters is an expression of our chosen simulation time of 20 kyrs in combination with the slow sedimentation rate of 10 cm kyr$^{-1}$.

With increasing SOM lability, the SOC profiles get more pronounced with highest concentrations in the upper layers and lowest concentrations in the lower part of the soil (Figure A11). With increasing thickness of the active layer, less SOC gets

incorporated into permafrost. This decrease is a consequence of a longer transport distance to the permafrost table, and therefore more time for conditions favourable to decomposition.

The implemented transport scheme does not fully capture vertical SOC distributions as inferred from observations (like e.g. an increase with depth in SOC in the uppermost turbel soil profile, Harden et al., 2012). But the scheme allows capturing the general tendency of decreasing SOC contents with depth, especially the lower SOCC at the permafrost table as compared to

mean SOCC in the active layer (which determines the SOC transfer between permafrost and active layer carbon in our model, see next section)

The lability dependent decline in SOCC leads to a stronger fractionation of SOM into slow and fast cycling SOC, resulting in a higher share of more labile SOC under cold climate conditions as compared to more moderate climate conditions. E.g. the share of labile SOC getting incorporated into perennially frozen ground is negligible with the slow pool representing the

largest contribution to permafrost SOC build-up (Figure A11, upper panel), in contrast to much higher shares of labile components (Figure A11, lower panel).

### 7.7.2    Implementation of the process-based model of SOM transport into JSBACH

For calculating the transfer of SOC between perennially frozen and seasonally thawed pools in JSBACH, the SOC concentration at active layer depth $SOCC_{ALD}^{i}$ (see Eq. 1) is required for each lability class per year and grid cell. If soil

temperatures increase, the active layer is deepening (with the extent of this active layer deepening being simulated by JSBACH) and SOC is transferred from non-active to active carbon pools (Figure 1). Hereby we assume that SOCC in the perennially frozen pools can be approximated as constant with depth (see Figure A11). If soil temperatures are decreasing, the active layer shrinks and SOC is transferred from the active to the non-active pools.

Based on the process-based model, we determine for each lability class the ratio of SOCC at the permafrost table $SOCC_{ALD}^{i}$

to mean $SOCC^{i}$ within the active layer for each grid cell and year (for a given ALD). The functional dependence of $SOCC_{ALD}^{i}$ on active layer depth is inferred from equilibrium simulations with the process-based model and is shown in Figure

A12. By using linear approximations, we can determine the individual soil organic carbon concentrations $SOCC^i_{ALD}$ for any modelled active layer depth in JSBACH.

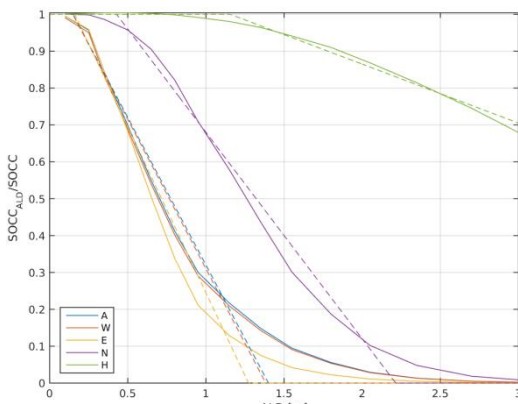

5  **Figure A12: Dependency of vertical soil carbon concentration at the permafrost table on active layer depth (ALD) as simulated by the process-based SOM transport model. The ratio of equilibrium SOCC at the active layer depth $SOCC^i_{ALD}$to mean seasonally thawed $SOCC^i$ is shown for individual soil carbon lability pools. Values smaller than one are indicative of SOCC declining with depth. Dotted lines illustrate linear approximations used for implementation into JSBACH. Curves are inferred for default parameters of the process-based SOM transport model (cryoturbation rate: 10 cm$^2$ yr$^{-1}$, sedimentation rate: 10 cm kyr$^{-1}$, litter**
10  **input described by grassland).**

Figure A12 illustrates the lability-dependent vertical decline in SOCC and shows that for active layer thicknesses larger than two meters the SOC transfer into permafrost in the process-based model is strongly dominated by the slow pool (green lines).

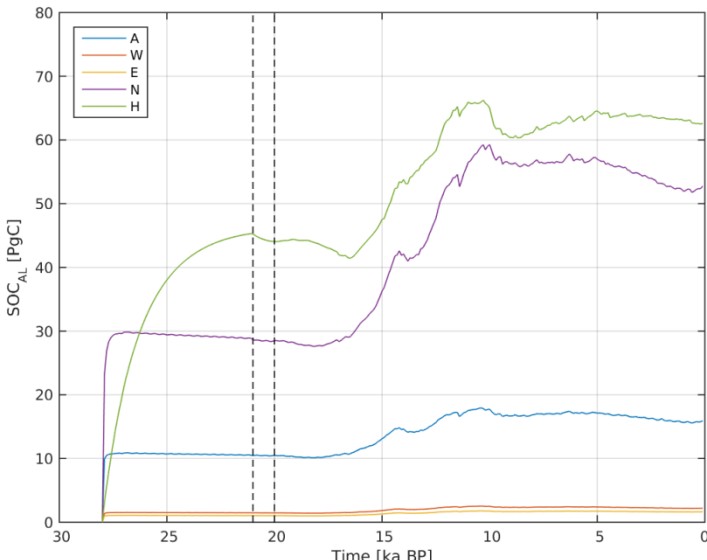

**Figure A 13: Simulation set-up for deglacial model runs. Shown is the spinup and fully transient phase of active layer SOC pools for all SOM lability classes (A, W, E, N, H) from the experiment initialization at 28 ka BP to 0 BP. The vertical dashed line at 21 ka BP illustrates the end of the SOC spinup phase, the vertical dashed line at 20 ka BP illustrates the end of the stationary LGM climate forcing. Individual SOC contributions were summed over all permafrost grid cells.**

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
