# Peer review of "Long-term deglacial permafrost carbon dynamics in MPI-ESM"

_Climate of the Past, 2018_

## Referee Comment (RC1) · Anonymous Referee #1 · 11 Jul 2018

This manuscript describes a new model formulation to represent permafrost carbon dynamics through glacial-deglacial timescales together with its sensitivity to several factors. It is indeed an important topic to tackle within our model capabilities since not many experiments have been performed in this field. Permafrost carbon dynamics are important components of the global biogeochemical cycles and there is much to discuss on their role in the recent deglaciation.

Authors have nicely presented their model development and findings. The manuscript is well organized and nicely written with easy to follow logical steps and relevant figures complemented with an extensive appendix section.

I support the results and development work shown by the authors, while some more focus is needed relating their findings to the overall conclusions in respect to the journal's

visions to distinguish from a solely model development study.

Please find my comments below.

Overall comments:

1. Constant depth organic layer insulation is obviously too strong for such an experiment. It would have been better to remove it completely since there is no way to constrain it at these timescales. Any plans to use a dynamic organic layer in a future work?

2. Is the solid green line - SOC(AL) in Fig. 7 and Fig. A12 show the same simulation results? I find the combined values in Fig. A12 slightly higher than the values of solid green line in Fig. 7.

3. Implications of constant soil depth should be further discussed. 11 vertical layers and a 40m depth limit, is this good enough to represent thermal diffusion over such long timescales? (no: Alexeev et al 2007, a 200 year simulation needs a 30m soil depth, so how much does a 20k year simulation need?!)

4. One the main issues is failing to capture the LGM pf extent. You have mentioned the related limitations of the model and forcing data. Other than the organic layer issue, you should also mention the more important snow insulation and how the model can create a much different soil thermal regime with an improper snow representation. There is a long list of literature on snow insulation, please include a small section in the manuscript.

5. The overall conclusion "... alternating phases of soil carbon accumulation and loss as an effect of dynamic changes in permafrost extent, active layer depths, soil litter input, and heterotrophic respiration." is too general and rather obvious. You have several sensitivity tests and and spatial analyses, please focus the conclusion on specific factors of uncertainty for different regions, and aim to quantify the reasons of model-data mismatches to these factors. Otherwise this is just a model development study

and misses the key element of improving our understanding of how to simulate past permafrost carbon dynamics in a better way.

Specific comments:

P1 L21-23: Do you mean the observational data reconstructions suggest a shift of permafrost coverage to southerly regions from glacial to interglacial times?

P1 L24: I couldn't see the actual comparisons to the model run without your new SOC transfer process. Please correct me or include relevant figures/tables to clarify this. This 'control' simulation is mentioned throughout the manuscript yet no result is shown from that experiment.

P3 L15: Crichton et al. (2016)'s work was already an ESM experiment. It would be useful to mention that you mean full and more complex ESM studies and not the EMICs.

P5 L2: please explicitly describe the symbols in the equation

P6 L11: Fig. A1?

P8 L27: Fig. A12 shows that the slow pool is not yet in equilibrium after 7000 years of spinup. Could this choice of spinup period be an effect for the underestimation of permafrost carbon stocks in your results? Please explain your justifications and implications of this choice of spinup time.

P13 L7: figure A2 not A1

P13 L15: not clear what you mean by underestimating glacial southward spread of permafrost. Are you talking about PI or LGM here? Would it be better to rephrase it as: deglacial spread of permafrost coverage to southern regions?

P15 L2: strong limiting factors (have to be plural)

P15 L3: closed → close

[Figure]

P17 L6-8: sentence repetition of P15 L3-5

P19 L10-14: you mention the SOC(pf) change depends on the region if ice cover change was prominent during deglaciation. It seems like in Eurasia, even though less affected by ice sheet retreat, shows more SOC(pf) accumulation during 10kyBP to PI in Fig. 7. Can you explain that?

---

## Referee Comment (RC2) · Anonymous Referee #2 · 17 Jul 2018

This manuscript presents a modelling effort to explore the changes in permafrost soil carbon stock during the last glacial-interglacial transition, the role of which may be important in the deglacial atmospheric $CO_2$ rising as hypothesized in previous studies. Transient deglacial experiments of complex land surface models are very lacking, so this study is a welcome contribution. A realistic representation of permafrost soil carbon dynamics requires vertically-resolved soil biogeochemistry in the model, whereas the starting version of JSBACH model in this study lacks a vertical structure for soil carbon. To tackle this problem, the authors added in the original soil carbon module (YASSO) an additional pool to represent perennially frozen soils, which exchanges carbon with the above active layer pool, while the exchange rate depends on changes in ALD and the carbon concentration at the boundary between active layer and permafrost as derived from another (stand-alone) vertically-resolved soil carbon build-up model. Model limitations were laid out and sensitivity experiments were conducted. The manuscript is nicely organized and well written. However, I do have some comments that need to be addressed before publication.

Overall comments:

(1) The vertical SOC profile generated by the SOM transport/build-up model assumes equilibrium conditions, as also mentioned in Section 2.5.4. The authors argued that the pools approach equilibrium at decadal to centennial time scales and thus would not bring much biases in a deglacial experiment. However, considering the slow processes of vertical transport (the diffusion and advection term and their coefficients in Equation A1), such relatively short equilibration time is not self-evident. Therefore, some plots to illustrate the time evolution of SOCC from zero to near-equilibrium, or from one equilibrium to another when climate and ALD have changed, are necessary.

Furthermore, the key relationship of $SOCC_{AL}{}^{i}$ vs. ALD (Figure A11) was implemented in JSBASH to infer the carbon transfer rate between the active layer pool and permafrost pool. However, I'm wondering if this relationship is robust, namely, the same ALD leads necessarily to a very similar $SOCC_{AL}/SOCC_{mean}$. I can imagine two soil sites with the same ALD but different seasonal soil temperature variations, say, one site with sandy soils which is very warm during summer and very cold during winter, the other one with insulating organic-rich soils which has small seasonal temperature amplitude. Will such differences change your $SOCC_{AL}$ – ALD relationship a lot?

Then, in Figure A11, the caption says the curves are for default parameters including "litter input described by grassland". Do you use these curves for forested grid cells as well, or do you have a separate set of curves for forests (in which I suppose the coefficients in Equation A2 to describe the vertical discretization of litter input would be different)?

(2) The climate forcing at PI and LGM were from the MPI-ESM_1.2T31 runs, which was compared against its CMIP5 version for the PI run. But how does it compare with some

re-analysis climate datasets (e.g. CRU-NCEP for the 1900s)? Some information about the climate bias by MPI-ESM_1.2T31 compared to observational data is important to interpret the bias in simulated vegetation productivity (Figure A4) and ALD (Figure A9) by offline JSBACH.

For the deglacial climate, it will be helpful if you can also plot the transient temperature (and perhaps precipitation), in addition to Figure A1 for the weight of interpolation.

(3) In many places in the manuscript you used "offline version of MPI-ESM", which is not very accurate because you ran in fact offline land surface model JSBACH for the deglaciation. MPI-ESM was run only for PI and LGM time slices, while the transient climate actually came from CLIMBER2.

Specific comments:

Figure 1: the name "Passive SOC" is misleading, as it suggests recalcitrant carbon pool which is not the case here. How about "Non-active SOC" or "Frozen SOC"?

Equation 1 (and other equations): please specify the unit of each variable. Besides, it is written "$SOCC_{ALD}$" in Equation 1 but "$SOCC_{AL}$" elsewhere.

P6, L2: the word "module" is a bit misleading, as it suggests something inside JSBACH, whereas it is a stand-alone model that provides the $SOCC_{AL}$ – ALD relationship, the latter then being implemented in JSBACH.

P8, L13: Note that on P28, L8 you mentioned that ice sheet grid cells were assigned zero precipitation so as to prevent vegetation and soil carbon accumulation. Then, why do you need another procedure here to remove SOC under ice sheets?

Equation 2: This equation was only used once (to initialize $SOC_{PF}$) after the first 7000 years of spin-up, and the evolution of $SOC_{PF}$ was then prognostically simulated using Equation 1, right? Please specify.

Section 3.2: For each experiment, please specify whether or not a different $SOCC_{AL}$/SOCC vs. ALD relationship was applied. I could expect, for example, some changes in the relationship when you increased the cryoturbation rate, while little change when you doubled litter input.

Besides, some information about the CPU hours for the transient deglacial runs will be very helpful.

P10: The configuration for each sensitivity test is described, but some justification for the choices of these parameter values is missing. For example, in L2P_LIT the litter input rate was doubled; is it because the simulated GPP during PI is about half of the observations (but note that Figure A4 also shows a too high GPP in North America)? In L2P_ALD the thermal conductivity of soil organic layer was reduced by half; is there some observational evidence to support this value, or is it just a simple way to compensate the bias in modelled soil temperature?

Figure 2: It is better to overlay the empirically-derived permafrost boundaries on the modelled maps, e.g. the IPA map for today and the Lindgren et al. 2016 for the LGM, to facilitate an evaluation.

P13, L7: Figure A1 → A2

Figure 3: In the legend, the ticks for the numbers do not match the segmentation of colors, which makes it hard to read the map. Please check all the figures that have a similar problem (e.g. Figure 5).

P17, L4: …of "glacial"? (this paragraph is discussing the low SOC bias for PI) Besides, Lines 6-8 duplicates a previous sentence.

Figure 6: It is interesting that permafrost area reaches maximum at 13 ka BP. How about adding a map of permafrost distribution for 13 ka to illustrate its changes compared to the LGM?

Figure 7: The caption says this is the total SOC summed for "near-surface permafrost from LGM to PI"; but permafrost extents (as well as unglaciated lands) are changing. Please specify which spatial area you have included in the summation.

P20, L4-6: This sentence does not read well, please rephrase.

P20, L23-25: What is the mechanism in the model that makes a lower vegetation productivity when ALD is shallower? Please specify.

Section 7.1: The spatial resolutions of MPI-ESM and CLIMBER2 are very different. How is this difference treated when you generate the transient climate forcing maps?

P24, L12: "lower GPP in North America and higher GPP in Eurasia" → "higher GPP in North America and lower GPP in Eurasia"

Figure A3: Is it possible to change the color scale so as to show the regional differences more clearly? A None-uniform color segmentation may be helpful in this case.

P26, L9: How does the temperature anomaly for the LGM compare with other PMIP3 models?

Equation A2: When the belowground litter flux is discretised along the depth, do you re-scale it to ensure carbon closure (especially when litter flux is cut by a shallow active layer depth)?

Figure A12: It will be helpful to include also $SOC_{PF}$. Besides, summation of all pools here seems to be higher than the green line in Figure 7?

---

## Author Comment (AC1) · 14 Oct 2018

We thank the referees for their constructive feedbacks which helped a lot to improve our manuscript. In the following we have listed a point-to-point reply to illustrate how we have accounted for all comments made.

**Referee 1:**

*Overall comments:*

1. *Constant depth organic layer insulation is obviously too strong for such an experiment. It would have been better to remove it completely since there is no way to constrain it at these timescales. Any plans to use a dynamic organic layer in a future work?*

We agree that organic layer insulation is a critical factor for subsurface soil temperatures and that it is hard to constrain this component over such long timescales. We had made experiments with variable organic layer insulation (by assuming a more shallow organic layer during glacial times than during interglacial times), but we found it hard to infer satisfactory model results for glacial and Holocene conditions with one consistent scheme as our dynamic organic layer insulation turned out to introduce a rather high sensitivity to subsurface soil temperatures. Therefore we decided to work with a constant layer scheme in this model study, while a more elaborate organic layer treatment needs further model development which is subject to current JSBACH development.

Under "Model limitation" in section 2.5.4. we had mentioned the aspect of constant layer depth, and now emphasize that this aspect needs improvement and is subject to current model development.

*2. Is the solid green line - SOC(AL) in Fig. 7 and Fig. A12 show the same simulation results? I find the combined values in Fig. A12 slightly higher than the values of solid green line in Fig. 7.*

Thanks for pointing to this inconsistency. Fig. A12 shows the time evolution of SOC pools which are not constrained to near-surface permafrost (as shown in Figure 7) but describes the full permafrost domain (including grid cells with ALD larger than 3m), and therefore suggest slightly larger values. We now specify the difference in summation of SOC pools in the legend of Figure 7 and Figure A12.

*3. Implications of constant soil depth should be further discussed. 11 vertical layers and a 40m depth limit, is this good enough to represent thermal diffusion over such long timescales? (no: Alexeev et al 2007, a 200 year simulation needs a 30m soil depth, so how much does a 20k year simulation need?!)*

We agree that our soil depth limit at 40m does not allow to fully capturing the thermal initeria provided by long-term glacial oscillations. Long-term millennial climate changes will especially affect permafrost thickness (i.e. the lower permafrost boundary). In our study we focus on changes in the upper permafrost boundary, i.e. the active layer thickness which is much less affected by long-term climate oscillations but rather by factors such as organic layer insulation or soil ice-content (factors which we discuss in the manuscript).

*4. One the main issues is failing to capture the LGM pf extent. You have mentioned the related limitations of the model and forcing data. Other than the organic layer issue, you should also mention the more important snow insulation and how the model can create a much different soil thermal regime with an improper snow representation. There is a long list of literature on snow insulation, please include a small section in the manuscript.*

We now have added relevant publications to underline the potential of soil temperature biases due to biased snow depth (and now stress this aspect in section 4.1.1. when discussing simulated LGM PF extent).

*5. The overall conclusion "... alternating phases of soil carbon accumulation and loss as an effect of dynamic changes in permafrost extent, active layer depths, soil litter input, and heterotrophic respiration." is too general and rather obvious. You have several sensitivity tests and and spatial analyses, please focus the conclusion on specific factors of uncertainty for different regions, and aim to quantify the reasons of model data mismatches to these factors. Otherwise this is just a model development study and misses the key element of improving our understanding of how to simulate past permafrost carbon dynamics in a better way.*

We now discuss continental-scale aspects of deglacial SOC evolution in more detail (in the results & discussion and conclusion sections, and we have added a further figure in the Appendix (see Figure below) which should help to improve our understanding which key factors drive deglacial SOC dynamics in permafrost regions. We especially discuss how changes in PF extent (due to warming-induced reduction in the area, or due to establishment of new PF following glacial retreat) and changes in deglacial NPP affect the SOC pools (see section 4.2.1). We now also discuss in more detail how climatology biases can translate into underestimating permafrost SOC pools via the coupling of active layer, soil moisture, and NPP.

[Figure]

Deglacial evolution of seasonally thawed (a) and perennially frozen (d) SOC in near-surface permafrostfrom LGM to PI. Panel (b) and (e) show deglacial evolution of NPP summed over near-surface permafrost grid cells, and permafrost extent. Panels (c) and (f) illustrate mean annual surface air temperature and active layer depth, which both were weighted over near-surface permafrost grid cells. Contribution from North America (light blue) and Eurasia (dark blue) are shown separately.

With regard to smaller-scale regional aspects we now discuss in section 4.1.4. conditions favourable for maximum SOC_PF accumulation in Siberia.

*Specific comments:*

*P1 L21-23: Do you mean the observational data reconstructions suggest a shift of permafrost coverage to southerly regions from glacial to interglacial times?*

We have now modified the wording to avoid misunderstanding.

*P1 L24: I couldn't see the actual comparisons to the model run without your new SOC transfer process. Please correct me or include relevant figures/tables to clarify this. This 'control' simulation is mentioned throughout the manuscript yet no result is shown from that experiment.*

We have now added a row in table 1 to describe the control experiment and added a row in table 2 to give numbers of the control experiment.

*P3 L15: Crichton et al. (2016)'s work was already an ESM experiment. It would be useful to mention that you mean full and more complex ESM studies and not the EMICs.*

Now accounted for

*P5 L2: please explicitly describe the symbols in the equation*

Now accounted for

*P6 L11: Fig. A1?*

We removed the wrong reference.

*P8 L27: Fig. A12 shows that the slow pool is not yet in equilibrium after 7000 years of spinup. Could this choice of spinup period be an effect for the underestimation of permafrost carbon stocks in your results? Please explain your justifications and implications of this choice of spinup time.*

We had run an experiment with 10 kyrs spinup which did not result in much increased permafrost carbon stocks. The choice of 7 kyrs has been a compromise between keeping computational time realistic and being not too far from equilibrium. For the experiment with increased turnover time for the slow pool (L2P_HDT), we increased the spinup time to 10 kyrs.

*P13 L7: figure A2 not A1*

Corrected for

*P13 L15: not clear what you mean by underestimating glacial southward spread of permafrost. Are you talking about PI or LGM here? Would it be better to rephrase it as: deglacial spread of permafrost coverage to southern regions?*

We here refer to LGM permafrost extent and adapted the sentence accordingly.

*P15 L2: strong limiting factors (have to be plural)*

Corrected for

*P15 L3: closed → close*

Corrected for

*P17 L6-8: sentence repetition of P15 L3-5*

*We removed the repetition.*

*P19 L10-14: you mention the SOC(pf) change depends on the region if ice cover change was prominent during deglaciation. It seems like in Eurasia, even though less affected by ice sheet retreat, shows more SOC(pf) accumulation during 10kyBP to PI in Fig. 7. Can you explain that?*

After 10 kyBP most of the ice sheet retreat has already been realized. A key factor for the SOC pools changes is a long-term shallowing of active layer depths after 10 kyrs BP which is generally larger in EA compared to NA grid cells (which finally increases SOC_PF stocks more strongly – see panel f in the above figure). Furthermore, the generally more shallow AL depths in EA result in more transport to PF due to a higher ratio of SOCC_AL/SOCC (see Fig. A11).

.

---

## Author Comment (AC2) · 14 Oct 2018

We thank the referees for their constructive feedbacks which helped a lot to improve our manuscript. In the following we have listed a point-to-point reply to illustrate how we have accounted for all comments made.

*Referee 2:*

*Overall comments:*

*(1) The vertical SOC profile generated by the SOM transport/build-up model assumes equilibrium conditions, as also mentioned in Section 2.5.4. The authors argued that the pools approach equilibrium at decadal to centennial time scales and thus would not bring much biases in a deglacial experiment. However, considering the slow processes of vertical transport (the diffusion and advection term and their coefficients in Equation A1), such relatively short equilibration time is not self-evident. Therefore, some plots to illustrate the time evolution of SOCC from zero to near-equilibrium, or from one equilibrium to another when climate and ALD have changed, are necessary.*

We agree that the full equilibration time is determined by the slow processes of diffusion and advection and results in equilibration on centennial to millennial timescales. Our statement of the pools equilibrating on decadal to centennial timescales applies for the SOC dynamics without vertical transport. We have now rephrased the corresponding section 2.5.4. For our simulations, the important aspect is the equilibration timescale of the ratio $R$ of SOCC_ALD/SOCC (which is illustrated below). The Figure illustrates the temporal evolution of $R$ along a spinup-period of 10 kyrs, followed by a fast warming or cooling scenario (1°C over 100yrs) for a shallow and a deep active layer setting. The transient excursions after 10 kyrs are larger than when inferred from a fully-dynamic model as the simplified model does not model transient litter input changes but adjusts litter input instantaneously to changing surface air temperatures and therefore overemphasizes the magnitude of the transient peaks. Yet the figure illustrates that the equilibration timescale is in the order of ~1000 yrs. We therefore now emphasize throughout the manuscript that the focus of our study is on capturing long-term (millennial scale SOC dynamics) (which was mentioned before in section 2.4, but not prominently enough to avoid misinterpretation of results).

[Figure]

**Transient evolution of the ratio SOCC$_{ALD}$/SOCC for all individual lability classes under fast warming and cooling for a shallow and a deep active layer site**

*Furthermore, the key relationship of SOCC$_{ALi}$ vs. ALD (Figure A11) was implemented in JSBASH to infer the carbon transfer rate between the active layer pool and permafrost pool. However, I'm wondering if this relationship is robust, namely, the same ALD leads necessarily to a very similar SOCC$_{AL}$/SOCC$_{mean}$. I can imagine two soil sites with the same ALD but different seasonal soil temperature variations, say, one site with sandy soils which is very warm during summer and very cold during winter, the other one with insulating organic-rich soils which has small seasonal temperature amplitude. Will such differences change your SOCC$_{AL}$ – ALD relationship a lot?*

No. We have tested different annual temperature cycles (in combination with modified MAGTs to infer the same ALD). While a smaller annual cycle favours more SOC build-up in the active layer (due to less respiration loss), the ratio $R$ *(SOCC$_{AL}$/SOCC$_{mean}$)* turned out to be rather stable. Differences in $R$ due to modified annual temperature cycles increase with deeper active layers. For our deep active layer setting (ALD=150cm), a reduction in the seasonal cycle by a factor of two (from 40°C to 20°C) results in deviations of R of typically a few percent. The largest deviation is seen for the intermediate N pool of 8% above its standard value (inferred from the seasonal cycle setting of 40°C as used in our manuscript).

*Then, in Figure A11, the caption says the curves are for default parameters including "litter input described by grassland". Do you use these curves for forested grid cells as well, or do you have a separate set of curves for forests (in which I suppose the coefficients in Equation A2 to describe the vertical discretization of litter input would be different)?*

No, we do not describe separate coefficients for southern PF grid cells which contain forests. As the vast majority of litter input in permafrost grid cells stems from C3 grass we do not use separate fit curves for an improved approximation of soil C profiles in mainly forested permafrost regions.

*(2) The climate forcing at PI and LGM were from the MPI-ESM_1.2T31 runs, which was compared against its CMIP5 version for the PI run. But how does it compare with some re-analysis climate*

*datasets (e.g. CRU-NCEP for the 1900s)? Some information about the climate bias by MPI-ESM_1.2T31 compared to observational data is important to interpret the bias in simulated vegetation productivity (Figure A4) and ALD (Figure A9) by offline JSBACH.*

We now refer to two recently submitted publications (Mauritsen et al. 2018, Mikolajewicz et al. 2018), which demonstrate the model performance of MPI-ESM1.2.

*For the deglacial climate, it will be helpful if you can also plot the transient temperature (and perhaps precipitation), in addition to Figure A1 for the weight of interpolation.*

We have prepared an additional Figure for the Appendix (see new figure in the response to reviewer 1) which shows the deglacial evolution of mean annual SAT weighted over the permafrost domains in Eurasia and North America (which we think allows to better interpret some aspects of deglacial SOC dynamics).

*(3) In many places in the manuscript you used "offline version of MPI-ESM", which is not very accurate because you ran in fact offline land surface model JSBACH for the deglaciation. MPI-ESM was run only for PI and LGM time slices, while the transient climate actually came from CLIMBER2.*

We have now modified the occurrences in the text were "offline" refers to MPI-ESM (and not to JSBACH).

**Specific comments:**

*Figure 1: the name "Passive SOC" is misleading, as it suggests recalcitrant carbon pool which is not the case here. How about "Non-active SOC" or "Frozen SOC"?*

We now have labeled the pools "Non-active".

*Equation 1 (and other equations): please specify the unit of each variable. Besides, it is written "$SOCC_{ALD}$" in Equation 1 but "$SOCC_{AL}$" elsewhere.*

Now accounted for.

*P6, L2: the word "module" is a bit misleading, as it suggests something inside JSBACH, whereas it is a stand-alone model that provides the $SOCC_{AL} - ALD$ relationship, the latter then being implemented in JSBACH.*

We now refer to "model".

*P8, L13: Note that on P28, L8 you mentioned that ice sheet grid cells were assigned zero precipitation so as to prevent vegetation and soil carbon accumulation. Then, why do you need another procedure here to remove SOC under ice sheets?*

We have now reformulated the corresponding sentence to avoid misinterpretation of a further SOC removal mechanism.

*Equation 2: This equation was only used once (to initialize $SOC_{PF}$) after the first 7000 years of spin-up, and the evolution of $SOC_{PF}$ was then prognostically simulated using Equation 1, right? Please specify.*

Equation 2 is used during the full spin-up period, but it is only the final timestep of the spin-up phase which fully determines $SOC_{PF}$ initialization (depending on how much SOC is initialized in the active layer). We now specify when $SOC_{PF}$ is calculated diagnostically and when prognostically.

*Section 3.2: For each experiment, please specify whether or not a different SOCC$_{AL}$/SOCC vs. ALD relationship was applied. I could expect, for example, some changes in the relationship when you increased the cryoturbation rate, while little change when you doubled litter input.*

Yes, the doubled cryoturbation rate affects the SOCC$_{AL}$/SOCC ratio, while this ratio was little or not affected by the other sensitivity experiments. Therefore we have used modified parameters for describing the SOCC$_{AL}$/SOCC ratio in experiment L2P_VMR and now mention this modified parameter setting in table 1.

*Besides, some information about the CPU hours for the transient deglacial runs will be very helpful.*

The model requires 16.43s per model year on 108 nodes of our high-performance machine, giving a total computation time requirement of 0.5 node-h/yr (we now give this information in section 3.2.).

*P10: The configuration for each sensitivity test is described, but some justification for the choices of these parameter values is missing. For example, in L2P_LIT the litter input rate was doubled; is it because the simulated GPP during PI is about half of the observations (but note that Figure A4 also shows a too high GPP in North America)? In L2P_ALD the thermal conductivity of soil organic layer was reduced by half; is there some observational evidence to support this value, or is it just a simple way to compensate the bias in modelled soil temperature?*

We have doubled litter input for compensating our inferred large GPP bias in Eurasia (of typically a factor of 2 low-bias). Model biases in GPP in North America are of opposite sign, but are also smaller and exert a smaller weight on total permafrost GPP given the much smaller PF extent in North America compared to Eurasia. The experiment was rather meant to test the sensitivity of simulated SOC accumulation due to increased litter input, rather than an attempt to minimize model biases between simulated and observed GPP.
The halving of thermal conductivity experiment was not based on observational evidence but attempted to improve the consistency of modelled with observed active layer depths (CALM, see Figure A9), and to demonstrate the sensitivity of SOC-buildup to simulated ALD.

*Figure 2: It is better to overlay the empirically-derived permafrost boundaries on the modelled maps, e.g. the IPA map for today and the Lindgren et al. 2016 for the LGM, to facilitate an evaluation.*

We refrained from overlaying empirically-derived permafrost boundaries given the difficulties of a coarse-resolution model in resolving observed or reconstructed smaller-scale permafrost occurrences. We rather decided to discuss aspects of model-data matches and mismatches in the text.

*P13, L7: Figure A1 → A2*
Corrected for.

*Figure 3: In the legend, the ticks for the numbers do not match the segmentation of colors, which makes it hard to read the map. Please check all the figures that have a similar problem (e.g. Figure 5).*

We have now solved the tick mismatch issue (Fig.3, Fig.5, Fig. A8)

*P17, L4: …of "glacial"? (this paragraph is discussing the low SOC bias for PI) Besides, Lines 6-8 duplicates a previous sentence.*

Now modified accordingly.

*Figure 6: It is interesting that permafrost area reaches maximum at 13 ka BP. How about adding a map of permafrost distribution for 13 ka to illustrate its changes compared to the LGM?*

We agree that it is an interesting (side) aspect that permafrost maximum is not close to LGM but occurs many millennia later. We think that this is a robust finding, but we are cautious about focusing on a time of maximum extent as this depends strongly on the skill of simulating deglacial permafrost spread. As we discuss in the manuscript, the comparison to reconstructions points to an underestimate of LGM permafrost extent (especially in southern regions). Therefore we do not want to over-interpret a PF maximum map for 13kyr.

*Figure 7: The caption says this is the total SOC summed for "near-surface permafrost from LGM to PI"; but permafrost extents (as well as unglaciated lands) are changing. Please specify which spatial area you have included in the summation.*

When summing up numbers, we account for changes in permafrost extent by performing the summation for each 100 year time step over all grid cells which are classified as permafrost for the given time interval. We now specify this aspect in the caption of Figure 7.

*P20, L4-6: This sentence does not read well, please rephrase.*

We have rephrased this sentence.

*P20, L23-25: What is the mechanism in the model that makes a lower vegetation productivity when ALD is shallower? Please specify.*

NPP is affected by ALD via soil moisture. We now specify this aspect in the manuscript as "Under LGM conditions, this SOC gain is compensated by simulating very shallow active layer depths in many grid cells which result in lower vegetation productivity in L2P_ALD compared to L2P as a consequence of modified soil moisture and soil water availability."

*Section 7.1: The spatial resolutions of MPI-ESM and CLIMBER2 are very different. How is this difference treated when you generate the transient climate forcing maps?*

Climber fields are first regridded to a 10x10° grid, taking the continental layout on the climber grid into account. They are then regridded to MPI-ESM's T31 resolution by performing a bilinear interpolation.

*P24, L12: "lower GPP in North America and higher GPP in Eurasia" → "higher GPP in North America and lower GPP in Eurasia"*

Corrected for

*Figure A3: Is it possible to change the color scale so as to show the regional differences more clearly? A None-uniform color segmentation may be helpful in this case.*

We have now re-plotted Figure A3 using a modified colour segment scaling to better emphasize regional differences (yet, given the graphical representation of the whole circum Arctic domain, we found it difficult to show regional aspects in much higher detail).

*P26, L9: How does the temperature anomaly for the LGM compare with other PMIP3 models?*

Simulated global LGM cooling is at the lower end of PMIP3 models (which range to a global mean surface air cooling up to ~5.5°C – see e.g. Schmidt et al., 2014,  Using palaeo-climate comparisons to constrain future projections in CMIP5, Climate of the Past). We now refer to this paper to put our simulated LGM anomaly into relation to PMIP3 model results.

*Equation A2: When the belowground litter flux is discretised along the depth, do you re-scale it to ensure carbon closure (especially when litter flux is cut by a shallow active layer depth)?*

Yes, we ensure carbon closure – also in the case of active layer less shallow than theoretical maximum root depth. We now mention this aspect in the Appendix.

*Figure A12: It will be helpful to include also SOCPF. Besides, summation of all pools here seems to be higher than the green line in Figure 7?*

We wanted to illustrate the dynamical spinup of active layer SOC pools. SOC_PF pools do not evolve dynamically during the spinup period. Dynamic changes in SOC_PF can be inferred from Figure 6.
Thanks for pointing to the inconsistency in SOC pool sizes. Fig. A12 shows the time evolution of SOC pools which are not constrained to near-surface permafrost (as shown in Figure 7) but describes the full permafrost domain (including grid cells with ALD larger than 3m), and therefore suggest slightly larger values. We now specify the difference in summation of SOC pools in the legend of Figure 7 and Figure A12.